# Detection and recognition of foreign objects in Pu-erh Sun-dried green tea using an improved YOLOv8 based on deep learning

**Houqiao Wang[1], Xiaoxue Guo[2], Shihao Zhang[2], Gongming Li [2]\*, Qiang Zhao[2]\*, Zejun Wang[1]\***

**1** College of Tea Science, Yunnan Agricultural University, Kunming, China, **2** College of Mechanical and Electrical Engineering, Wuhan Donghu University, Wuhan, China

\* 13797554490@163.com (GL); zhaoiqang1486wdy@wdu.edu.cn (QZ); wangzejun0529741X@163.com (ZW)

**Data Availability Statement:** The data presented in this study are available on request from the corresponding author.

## Abstract

The quality and safety of tea food production is of paramount importance. In traditional processing techniques, there is a risk of small foreign objects being mixed into Pu-erh sun-dried green tea, which directly affects the quality and safety of the food. To rapidly detect and accurately identify these small foreign objects in Pu-erh sun-dried green tea, this study proposes an improved YOLOv8 network model for foreign object detection. The method employs an MPDIoU optimized loss function to enhance target detection performance, thereby increasing the model's precision in targeting. It incorporates the EfficientDet high-efficiency target detection network architecture module, which utilizes compound scale-centered anchor boxes and an adaptive feature pyramid to achieve efficient detection of targets of various sizes. The BiFormer bidirectional attention mechanism is introduced, allowing the model to consider both forward and backward dependencies in sequence data, significantly enhancing the model's understanding of the context of targets in images. The model is further integrated with sliced auxiliary super-inference technology and YOLOv8, which subdivides the image and conducts in-depth analysis of local features, significantly improving the model's recognition accuracy and robustness for small targets and multi-scale objects. Experimental results demonstrate that, compared to the original YOLOv8 model, the improved model has seen increases of 4.50% in Precision, 5.30% in Recall, 3.63% in mAP, and 4.9% in F1 score. When compared with the YOLOv7, YOLOv5, Faster-RCNN, and SSD network models, its accuracy has improved by 3.92%, 7.26%, 14.03%, and 11.30%, respectively. This research provides new technological means for the intelligent transformation of automated color sorters, foreign object detection equipment, and intelligent sorting systems in the high-quality production of Yunnan Pu-erh sun-dried green tea. It also provides strong technical support for the automation and intelligent development of the tea industry.

**Funding:** This research was supported by the Scientific Research Fund Project of the Department of Education of Yunnan Province (No. 2022Y282) and the "Intelligent Storage of Puer Tea based on LoRa Technology" project (No. 2022Y231). The research funder (Gongming Li) had no role in the design, data collection and analysis, decision to publish, or preparation of the manuscript. However, they did make the page charge payment after the publication and participated in the project's investigation, methodology, and provision of resources. Our research is entirely based on scientific principles and independent academic judgment.

**Competing interests:** The authors have declared that no competing interests exist.

## Instruction

Yunnan's favorable geography and climate have given birth to Pu-erh tea, a jewel in China's and the world's tea culture [1,2]. As the cornerstone of Pu-erh tea's distinctive flavor, the production process of sun-dried green tea is vital for quality control [3]. However, the modern industrial production of Pu-erh sun-dried green tea still primarily relies on traditional manual trash removal [4]. Despite the precision advantages of manual sorting, it is labor-intensive, inefficient, and prone to subjective biases, making it ill-suited for the high efficiency and standardization demands of modern mass production [5]. Currently, the tea industry's level of mechanization and automation is at an early stage, lacking in efficient automated detection and sorting technologies, which have become a key bottleneck for technological progress and quality assurance in the Pu-erh sun-dried green tea industry [6].

With rising food safety standards and increasing consumer demand for high-quality tea, the application of automated and intelligent foreign object detection technology in the production of sun-dried green tea has become particularly urgent [7,8]. Existing detection technologies are significantly inadequate for small object detection, where small foreign objects in sun-dried green tea, due to their small size and inconspicuous features, coupled with the complexity of the production environment, pose challenges to the accuracy and robustness of foreign object detection [9,10]. Currently, there is no specialized recognition model for the detection of small foreign objects in sun-dried green tea, and this research gap urgently needs to be filled [11].

The rapid development of deep learning technology has brought new opportunities to the field of agricultural product quality detection [12]. The YOLO (You Only Look Once) series of algorithms have been widely applied in defect detection [13], quality grading [14], safety detection [15], and foreign object recognition [16] of agricultural products. Fan et al. [17] developed a deep learning architecture based on convolutional neural networks for online detection of apple defects. Deng et al. [18] proposed an automatic grading system for carrots based on computer vision and deep learning, using the lightweight deep learning model to achieve surface quality detection of carrots. Wu et al. [19] improved the YOLOv7 network, achieving an average detection accuracy of 91.12% for tea buds through RGB-D multimodal feature fusion. Wang et al. [20] proposed an improved TBC-YOLOv7 algorithm for grading detection of tea buds, enhancing the connection of global feature information by integrating a context-based Transformer module, with an average accuracy of 90%, an overall average accuracy of 87.5%, and a 3.4% improvement over the original YOLOv7. Li et al. [21] studied and proposed a lightweight improved YOLOv5s model for detecting dragon fruit in daytime and nighttime lighting environments. The model achieved an average accuracy rate of 97.80% and a frame rate of 139 FPS in the GPU environment, with a model size of only 2.5 MB. The model was successfully deployed on Android devices, achieving real-time dragon fruit detection. Bello et al. [22] studied a drone vision system based on Mask YOLOv7 for automatic detection and counting of cattle. The system achieved detection accuracies of 93% and 95% in controlled and uncontrolled environments, respectively, demonstrating its potential in automatic monitoring and reporting in animal husbandry. Meng et al. [23] proposed a model based on STCNN (Spatio-Temporal Convolutional Neural Networks) for unmanned pineapple harvesting. The model used the Shifted Window Transformer to fuse regional convolutional neural networks, achieving high-precision detection of pineapples. In complex field environments, the model achieved a detection accuracy of 92.54% and an average inference time of 0.163 seconds. Tahir et al. [24] proposed a model that integrates the Swin Transformer and CBAM (Convolutional Block Attention Module) modules, as well as Soft-NMS (Non-Maximum Suppression) technology, to improve the detection performance of small targets and occluded objects. On the

VisDrone2019 dataset, the average detection accuracy increased by 4.8% compared to YOLOv8s. The aforementioned research results not only prove the application potential of the YOLO series algorithms in the field of agricultural product quality detection, but also demonstrate the continuously improvable characteristics of the series, with significant advantages in improving detection accuracy, efficiency, and automation levels [25,26]. Therefore, the YOLO series algorithms have tremendous potential and value in achieving automation and intelligence in agricultural product processing through the detection and identification of small foreign objects in Pu-erh raw green tea.

This study aims to develop an improved YOLOv8-based model for detecting and identifying small foreign objects in Pu-erh sun-dried green tea. It incorporates the MPDIoU loss function to enhance detection accuracy and target precision [27], integrates the EfficientDet module for efficient size-target detection [28], and introduces the BiFormer mechanism for enhanced context understanding in images [29]. By merging slice-assisted super reasoning with YOLOv8, the model achieves greater accuracy and robustness in recognizing small and multi-scale objects [30]. These advancements offer new technical solutions for the intelligent transformation of automated sorting machines, foreign object detectors, and smart sorting systems in the high-quality production of Yunnan Pu-erh tea, strongly supporting the industry's move towards automation and intelligent development.

## Materials and methods

### Sample preparation and foreign object analysis

The processing workflow of Pu-erh sun-dried green tea, depicted in Fig 1A with its unique standards, is fraught with uncertainty. To ensure representativeness and scientific rigor, this study employed random sampling to capture the diversity of this traditional craft. Between March and April 2024, 75 Pu-erh sun-dried green tea samples were randomly collected from the tea horse ancient town tea trading hub in Simao District, Pu-erh City, Yunnan Province (22.47˚ N latitude, 100.58˚ E longitude), using the five-point sampling method [31] as shown in Fig 1B. All collected samples were mixed and homogenized to form the Pu-erh Sun-Dried Green Tea specimens required for this study's image acquisition. During the processing of Pu-erh sun-dried green tea, various foreign objects may be encountered. As shown in Fig 1C, this study identified four common types of foreign objects: Grain, Melon seed shells, Small twigs, and Bamboo chips. These objects differ significantly in material, size, and color, and their presence affects the quality and safety of the tea.

To simulate potential foreign matter encountered in actual processing, this study mixed Pu-erh sun-dried Green Tea with four common contaminants. In accordance with stringent quality requirements, the blended samples were evenly divided into 200 groups, each representing a possible foreign matter scenario in the processing of Pu-erh sun-dried green tea. These samples will be utilized for subsequent image collection and photography, providing a rich foundation of image data for this research.

### Image acquisition

In this study, to ensure high-quality capture of Pu-erh sun-dried green tea mixed with foreign matter images, a 20-megapixel MV-CE200-11UC economical area scan camera with a Sony IMX183 sensor was used, designed for capturing vibrant CMOS images. This industrial camera, with a maximum frame rate of 19.2fps, ensures the smoothness and real-time nature of the image acquisition process, meeting the study's need for high-speed imaging. To further enhance image clarity and accuracy, a 35 mm focal length MVL-LF3528M-F industrial lens was equipped. With a low optical distortion rate of 0.40% and an F-Mount interface, this lens

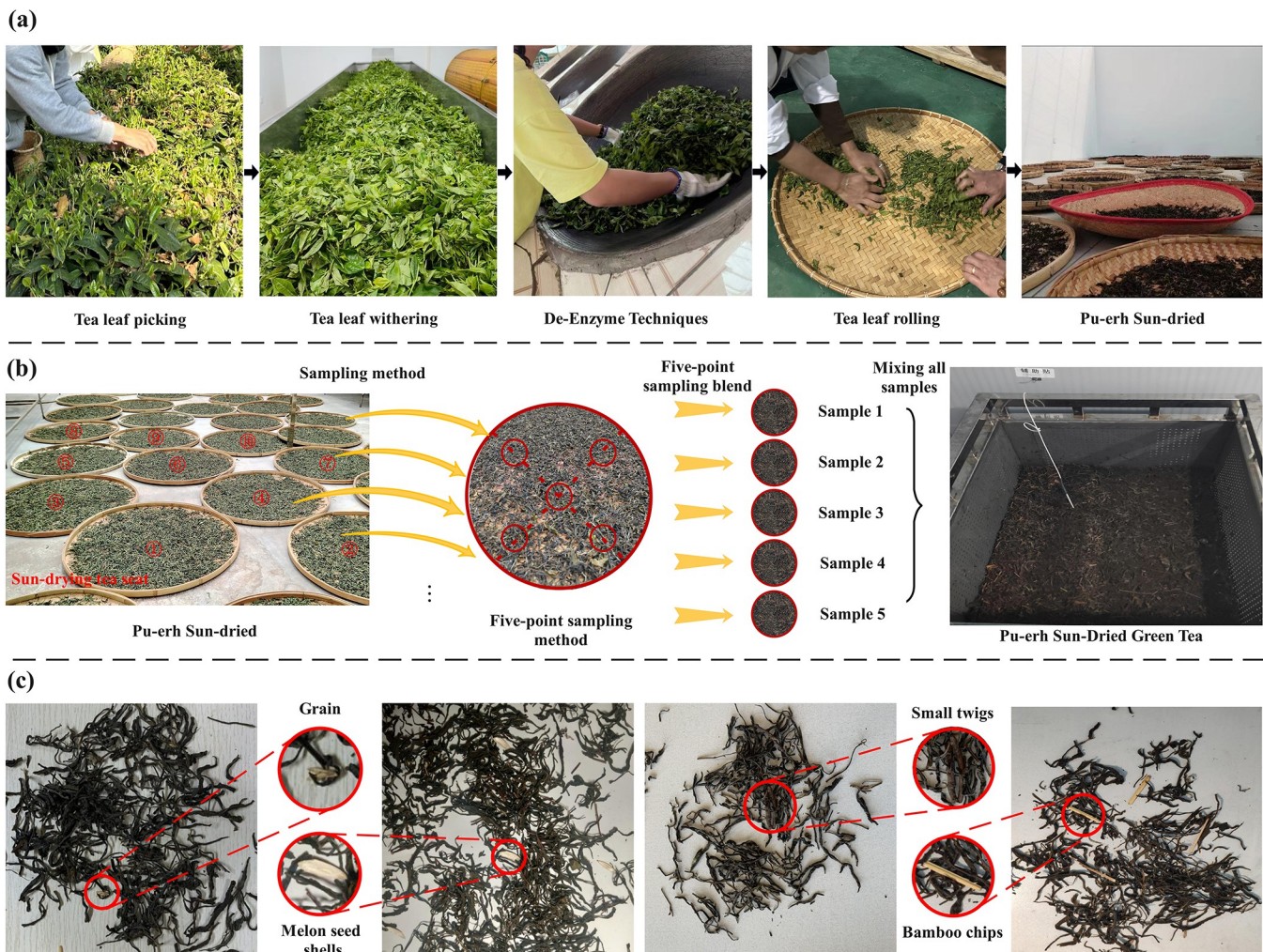

**Fig 1. Sample data set preparation.**

provides a robust guarantee for image quality. The low distortion rate minimizes edge distortion, ensuring authenticity and precision in the images.

## Image preprocessing and dataset division

This study collected a total of 938 images of Pu-erh sun-dried green tea mixed with Grain, Melon seed shells, Small twigs, and Bamboo chips. To enhance the dataset's quality and representativeness, 856 images were rigorously selected as the original foundational dataset. To address the potential negative impact of sample imbalance on the model's generalization, data augmentation techniques as shown in Fig 2 [32] were employed, including brightness adjustment [33], contrast enhancement [34], and flipping [35], to expand the dataset of Pu-erh sun-dried green tea with foreign objects. Ultimately, a dataset of 5992 images was obtained, effectively preventing overfitting during training and significantly enhancing the model's generalization, accuracy, and reliability in practical applications.

Labels were manually annotated using the Make Sense tool on the image dataset, with target bounding boxes centered on foreign objects, and label files created in txt and xml formats. As shown in Fig 3, the dataset labels include specific information such as the quantity of foreign

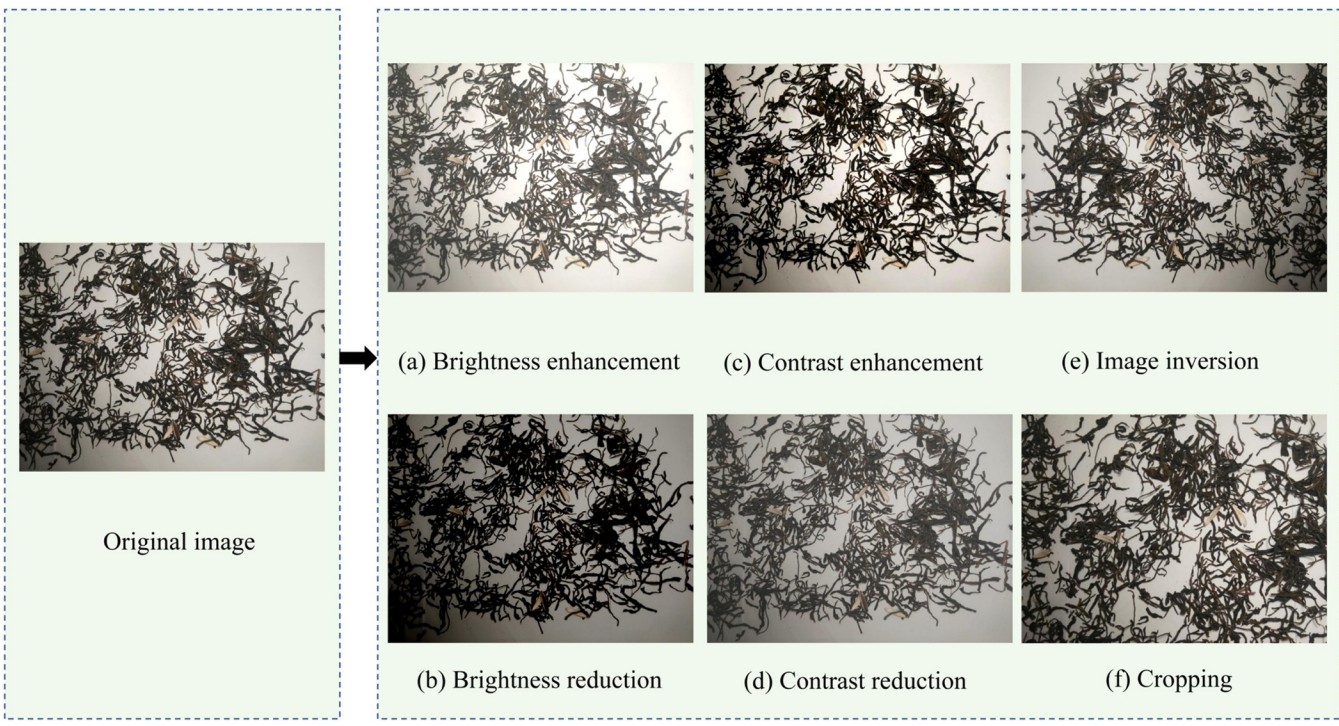

**Fig 2. Data enhancement processing.**

object classifications, the dimensions and distribution of bounding boxes, and the position coordinates of each foreign object center relative to the entire image. Additionally, it covers the aspect ratio of the target foreign objects relative to the entire image dataset. Labels A, B, C, and D represent Grain, Melon seed shells, Small twigs, and Bamboo chips, respectively.

In this study, to ensure the accuracy and generalization of model evaluation, a 5-fold cross-validation method [36] as depicted in Fig 4 was utilized. Initially, the dataset was randomly divided into an 80% training set and a 20% test set. Subsequently, the training set was further divided into 5 parts, alternately serving as sub-training and sub-validation sets in a 4:1 ratio over 5 cycles. Ultimately, the model's comprehensive performance metrics were derived from the average results of the 5 evaluations. This approach effectively prevents overfitting and ensures the stability of the assessment outcomes.

### Improvements to the YOLOv8 network

YOLOv8 is a deep learning model designed for real-time object detection, inheriting the core strengths of the YOLO architecture, which enables fast and accurate detection and localization by processing the entire image in a single pass within a unified neural network framework [37]. YOLOv8 innovates by incorporating advanced feature extraction mechanisms and optimized training strategies, significantly enhancing the model's performance in detection accuracy and speed [38]. This study improves upon the YOLOv8 framework by adopting the MPDIoU loss function to enhance detection performance and the model's precision in targeting; integrating the EfficientDet module for efficient detection of objects of various sizes through composite scale-anchor boxes and adaptive feature pyramids; introducing the BiFormer bidirectional attention mechanism to allow the model to consider both forward and backward dependencies in sequence data, thereby significantly enhancing the model's

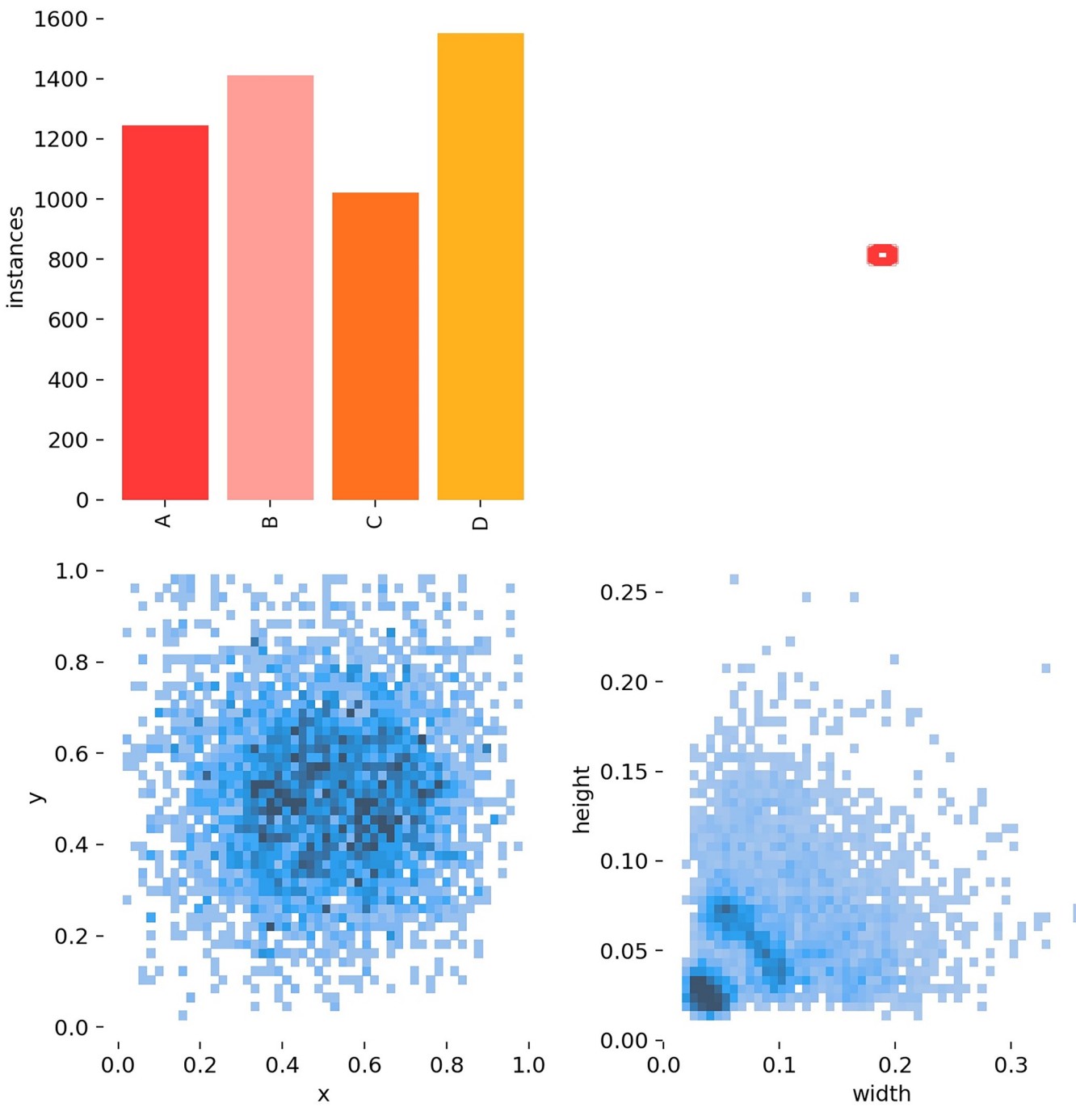

**Fig 3. Data information for the data set label.**

understanding of target context in images; and applying slice-assisted super reasoning technology in conjunction with YOLOv8 for algorithmic integration, which subdivides the image for in-depth local feature analysis, significantly improving the model's recognition accuracy and robustness for small and multi-scale objects. The structure of the improved YOLOv8 model is shown in Fig 5.

**Improvement of loss function.** The YOLOv8 model's core lies in its Boundary Bounding Regression (BBR) module, responsible for precisely determining target locations [39]. The loss

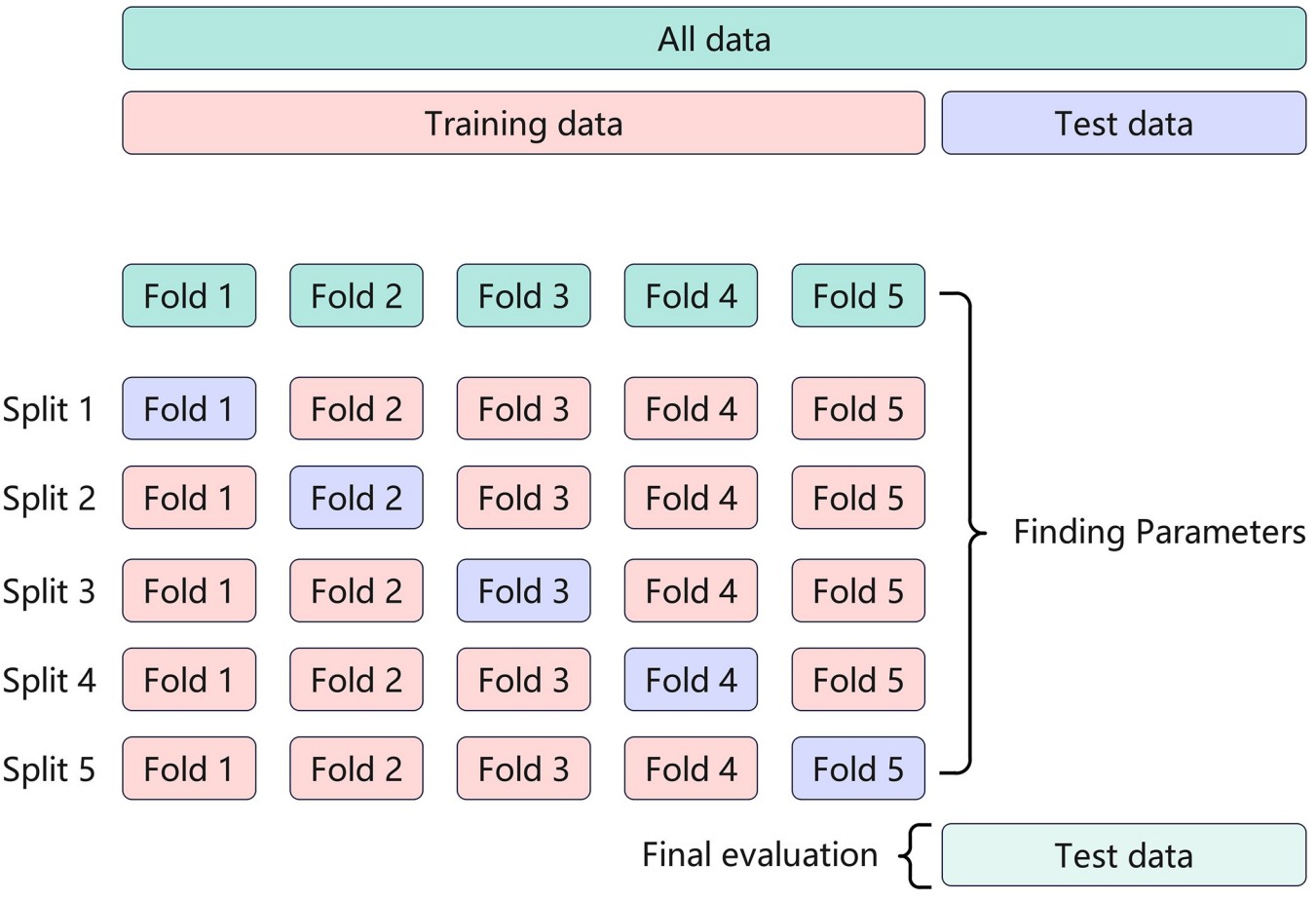

**Fig 4. 5-Fold cross validation.**

function within BBR is crucial, directly affecting the model's detection performance. Initially, YOLOv8 employed the CIoU loss function, which significantly improved detection box convergence by considering overlap area, center point distance, and aspect ratio consistency [40]. However, the CIoU loss function's effectiveness relies on high-quality experimental data in the training dataset. When the dataset includes low-quality data, the regression performance may be suboptimal [41]. Particularly in image datasets like Pu-erh sun-dried green tea, where small foreign objects are densely structured and bounding boxes have high overlap, the CIoU loss function may decline in convergence and accuracy.

To address this challenge, this study adopts the MPDIoU loss function [27], with structural parameters shown in Fig 6. The MPDIoU loss function is designed to overcome the limitations of traditional CIoU loss functions when dealing with highly overlapping bounding boxes. By optimizing the algorithm, it enhances the model's adaptability to dense targets and complex scenes, significantly improving detection accuracy and robustness. The computation formula is presented as follows.

$$d_1^2 = \left(x_2^A - x_1^A\right)^2 + \left(y_2^A - y_1^A\right)^2 \tag{1}$$

$$d_2^2 = \left(x_2^B - x_1^B\right)^2 + \left(y_2^B - y_1^B\right)^2 \tag{2}$$

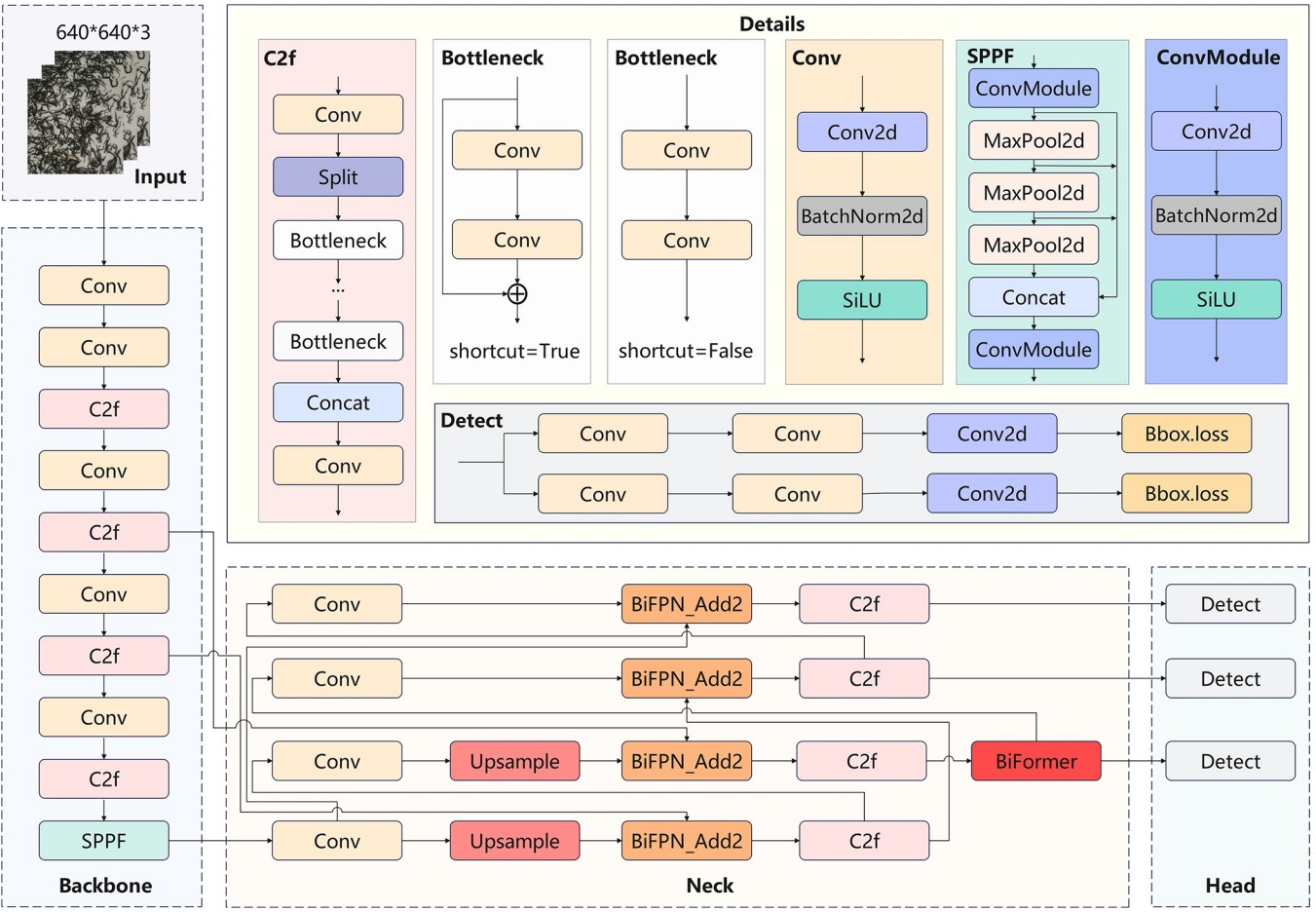

**Fig 5. Improved YOLOv8 model structure.**

$$MPDIoU = \frac{A \cap B}{A \cup B} - \frac{d_1^2}{W^2 + H^2} - \frac{d_2^2}{W^2 + H^2} \qquad (3)$$

Where, $(x_1^A, y_1^A)$ and $(x_1^B, y_1^B)$ represent the top-left and bottom-right coordinates of the actual bounding box, respectively; $(x_2^A, y_2^A)$ and $(x_2^B, y_2^B)$ represent the top-left and bottom-right coordinates of the predicted bounding box, respectively; $W$ and $H$ denote the width and height of the smallest enclosing rectangle covering both the predicted and actual bounding boxes; $d_1^2$ and $d_2^2$ represent the squares of the Euclidean distances between the top-left and bottom-right corners of the true and predicted boxes, respectively.

The definition of the MPDIoU loss is expressed as follows in Eq (4):

$$L_{Inner-IoU} = 1 - IoU^{Inner} \qquad (4)$$

The loss function directly minimizes the distance between the top-left and bottom-right points of the predicted and actual bounding boxes. This means the MPDIoU loss function is comprehensive and simplifies calculations. It also streamlines the comparison of similarity between bounding boxes, making it suitable for both overlapping and non-overlapping box regression. This indicates that the MPDIoU loss function can effectively improve the training of bounding

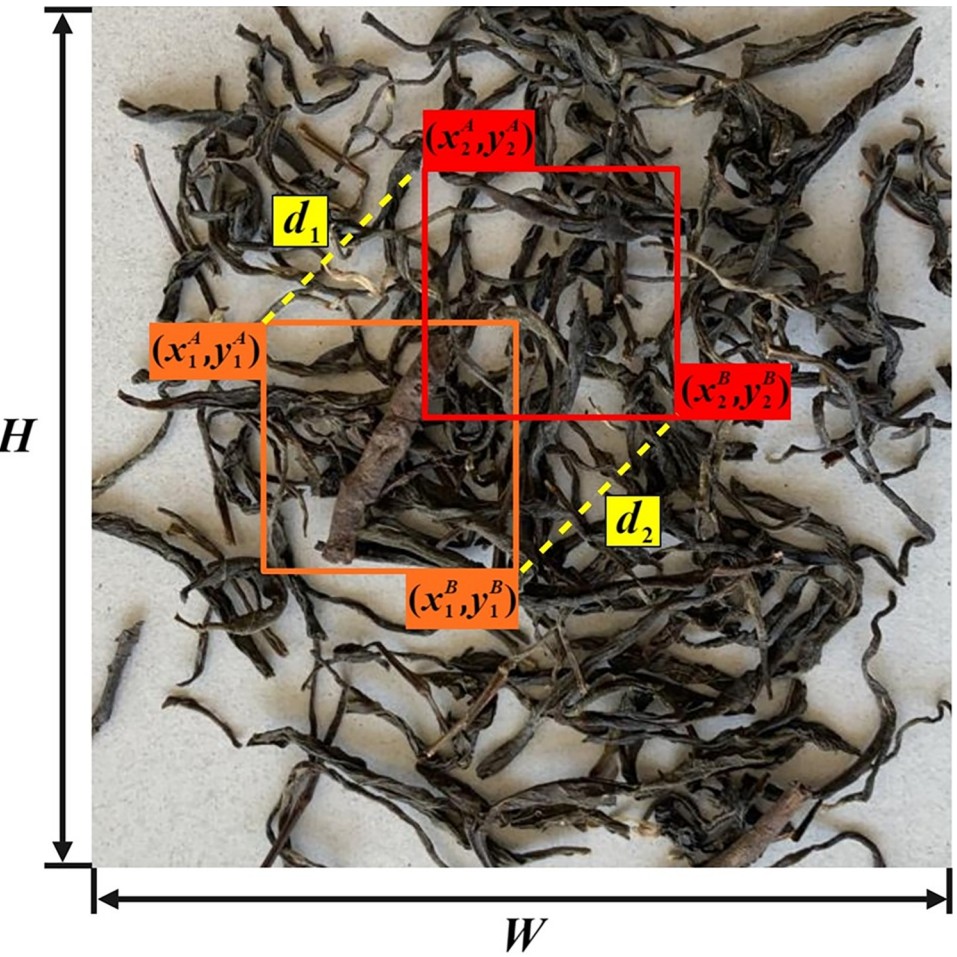

**Fig 6. MPDIoU structure parameter diagram.**

box regression for small foreign objects in Pu-erh sun-dried green tea, enhancing convergence speed and regression accuracy.

**EfficientDet.** The EfficientDet algorithm is a one-stage object detection network that extends the compound scaling concept of EfficientNet, clearly formalizing the network architecture into a scalable framework. It balances the precision of object detection while considering the model's detection speed [28].

The EfficientDet algorithm implementation, as shown in Fig 7, is divided into three stages: 1) The backbone EfficientNet extracts features using Neural Architecture Search and Compound Scaling Method; 2) The BiFPN integrates cross-layer features to enhance detection; 3) Classification and regression networks predict image categories and object locations. EfficientNet's initial and final layers consist of a single Conv, forming standard convolutional layers processed with normalization and activation functions, with kernel sizes of 3x3 and 1x1, respectively. All intermediate layers are composed of repeated Mobile Inverted Bottleneck Convolution (MBConv) stacks. The MBConv module initially increases dimensions with a 1x1 convolution, followed by a Depthwise Separable Convolution (DSC) network with either 3x3 or 5x5 kernels, and a Squeeze and Excitation Network (SENet), then reduces dimensions with a final 1x1 convolution, as depicted in Fig 8.

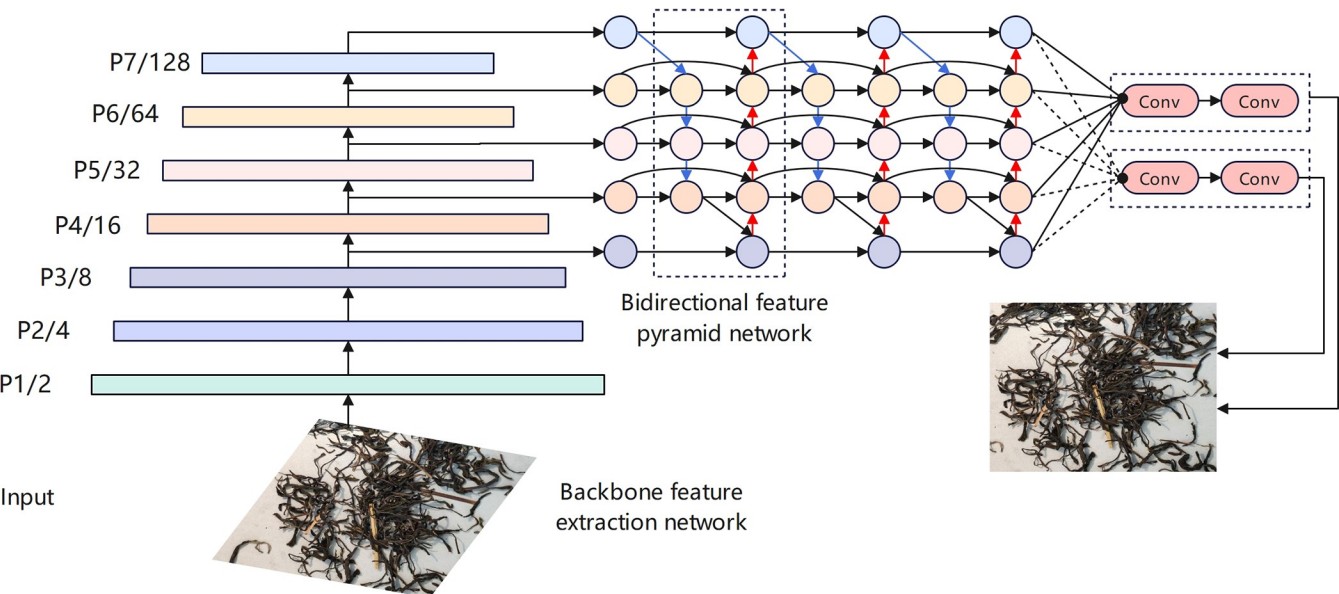

**Fig 7. EfficientDet network structure diagram.**

BiFPN is the core of the EfficientDet network, introducing bidirectional cross-scale connections and utilizing the allocation of training weights through fast normalization, providing a new solution for effectively representing and processing multi-scale features. Although the traditional Feature Pyramid Network (FPN) [42] can effectively handle multi-scale features, there is an issue where low-level information is lost when it passes through multiple layers of the network to reach higher layers due to the long path, as shown in Fig 9A. To reduce the loss of low-level information, the Path Aggregation Network (PANet) [43] adds a bottom-up path to the top-down structure of FPN and captures feature information at all levels through adaptive functional pooling, as shown in Fig 9B.

Due to the extensive memory requirements for the complex network computations in PANet, which leads to prolonged model inference times, BiFPN has been improved upon the basis of PANet. It incorporates bidirectional cross-scale connections by adding an additional connection path between the original input and output nodes at the same level, and by stacking the same feature layers multiple times. This approach achieves a balance between detection accuracy and computational speed, as shown in the network structure depicted in Fig 10.

**BiFormer.** The Self-Attention mechanism, renowned for capturing long-range dependencies, has become a pivotal technique in the field of object detection [44]. However, it also introduces significant memory consumption and high computational costs. To address these challenges, researchers have incorporated various manually designed sparse attention patterns

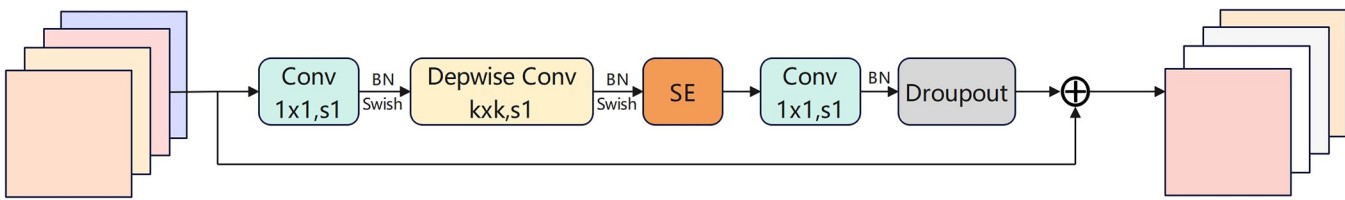

**Fig 8. EfficientDet network structure diagram.**

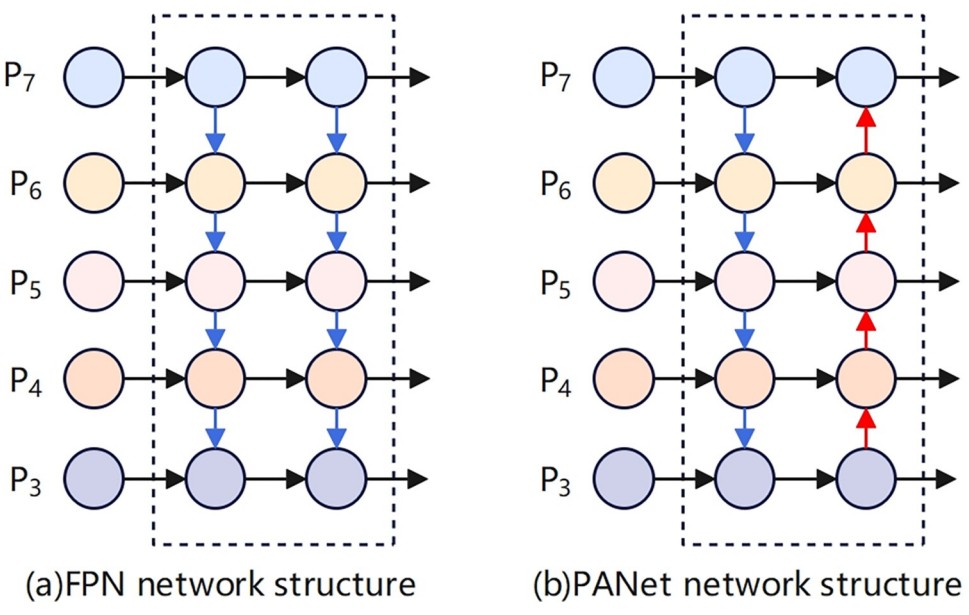

**Fig 9. Network structure of FPN and PANet.**

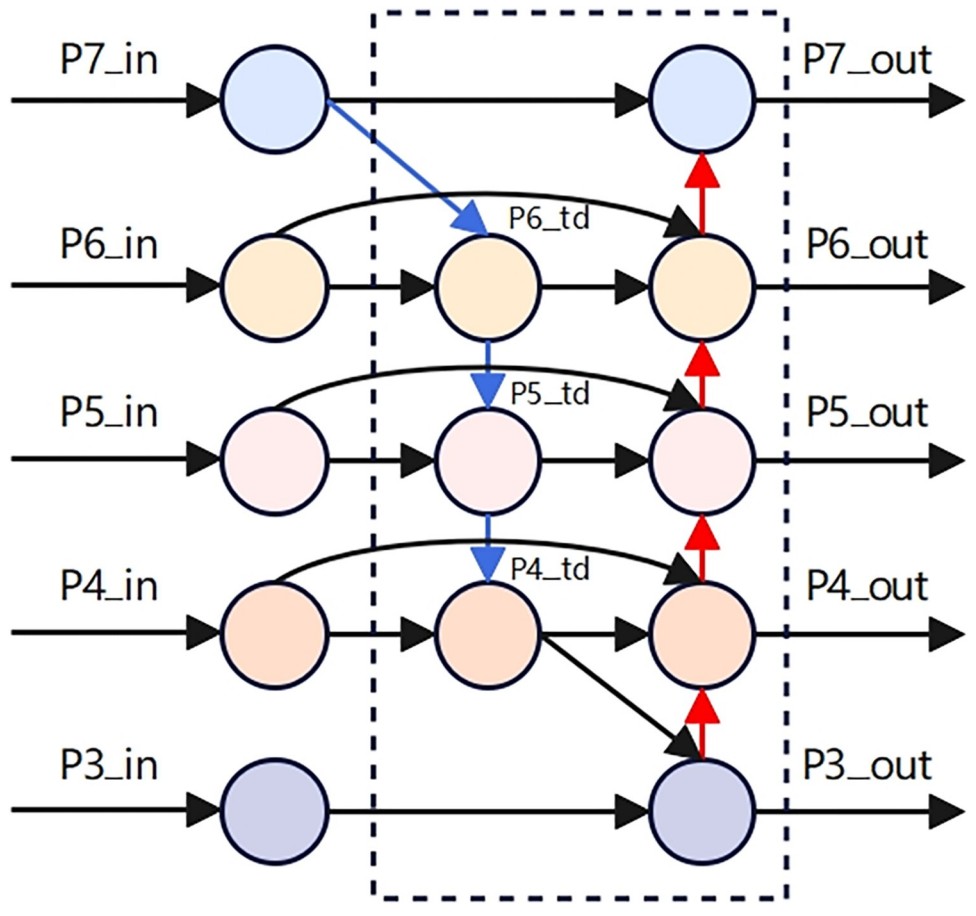

**Fig 10. BiFPN network structure.**

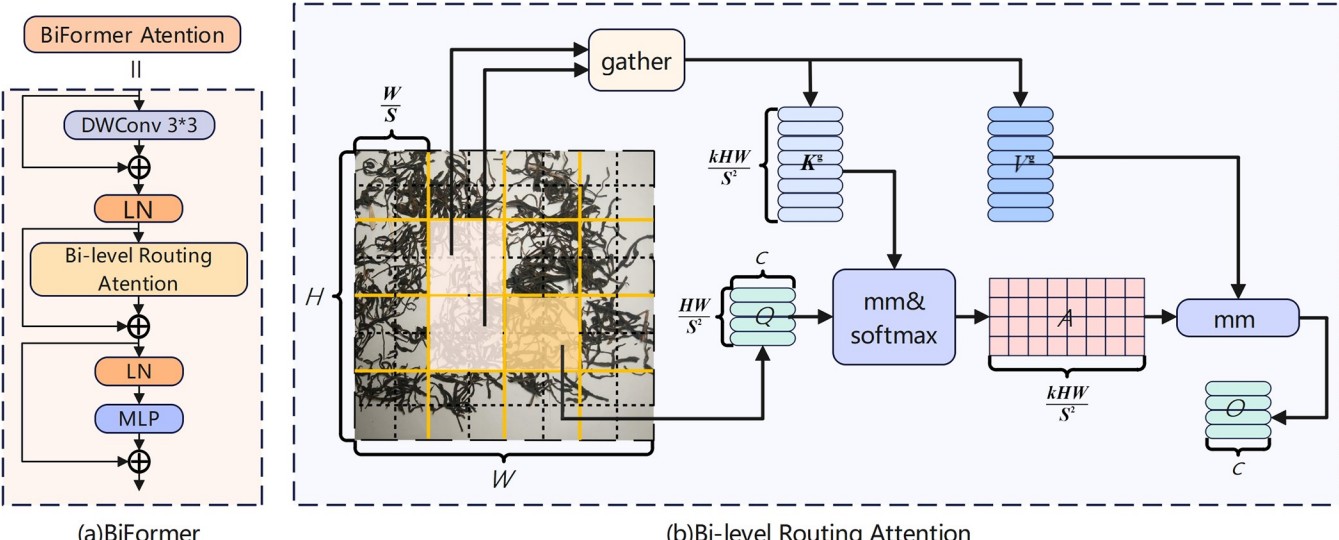

**Fig 11. Overall structure of BiFormer.**

to reduce model complexity [45,46]. While these methods alleviate computational pressure, they still fall short in fully capturing long-range relationships. Zhu et al. [29] proposed an innovative dual-routing attention mechanism, Bi-level Routing Attention (BRA), which employs a two-tier routing strategy to more effectively address the issue of capturing long-range dependencies. Based on the BRA core component, this study adopts the BifFormer general visual network architecture. The central design concept of BifFormer is to preliminarily filter out irrelevant key-value pairs at a coarse regional level using the BRA module, followed by applying a refined token-to-token attention mechanism in the remaining routing areas. This strategy not only endows the model with adaptability but also significantly enhances computational efficiency and substantially reduces memory usage. Consequently, BifFormer inherits the advantages of the Transformer model while achieving more flexible content awareness and computational resource allocation, as shown in the overall structure of the BifFormer model in Fig 11.

Assuming an input two-dimensional feature map $X \in \mathbb{R}^{H \times W \times C}$, it is divided into $S \times S$ non-overlapping regions, each $\frac{H \times W}{S^2}$ region containing a set of feature vectors. By reshaping $X$, it is possible to perform region partitioning for $X^r \in \mathbb{R}^{S^2 \times \frac{HW}{S^2} \times C}$, and after linear mapping, obtain the tensor $Q, K, V$, with the linear projection described as follows:

$$Q = X^r W^q, \; K = X^r W^k, \; V = X^r W^v \tag{5}$$

Where, $W^q W^k W^v \in \mathbb{R}^{C \times C}$ represents the projection weights for the tensor $Q, K, V$.

Based on a directed graph, routing from region to region is implemented. At the regional level, by calculating the average of $Q$ and $K$, the regional $Q^r K^r \in \mathbb{R}^{S^2 \times C}$ is obtained, and by performing matrix multiplication on $Q^r K^r$, an adjacency matrix for the region-to-region affinity graph is constructed:

$$A^r = Q^r (K^r)^T \tag{6}$$

Each entry in $A^r$ represents the semantic relevance between two regions. By retaining the top k most relevant connections for each region and pruning the affinity graph, the routing index

matrix $I^r$ is obtained:

$$I^r = topkIndex(A^r) \tag{7}$$

Where, the $i$-th row of $I^r$ contains the k most relevant indices for the $i$-th region.

The token-to-token attention uses the region-to-region routing index matrix $I^r$ to apply fine-grained token-to-token attention within the selected k routing regions. Since these routing regions may be scattered throughout the entire feature map, the key $K^g$ and value $V^g$ tensors are gathered:

$$K^g = gather(K, I^r), V^g = gather(V, I^r) \tag{8}$$

The attention mechanism is applied to the collected key-value pairs to compute the output $O$.

$$O = Attention(Q, K^g, V^g) + LCE(V) \tag{9}$$

Where, $LCE(V)$ is a local context enhancement term that effectively enhances the representational capability of local contextual features while ensuring that computational efficiency is not compromised. This design provides more precise and rich feature representation for subsequent foreign object detection tasks.

**Slice-assisted super inference algorithm integration.**   The Slice-assisted super inference involves slicing the input detection image and applying object detection to each slice [47], as illustrated in the steps shown in Fig 12.Assuming the input image size is $w \times h$, the overlap size $S$ and the movement distance $x$ are calculated based on the pre-set slice size $m \times m$ and overlap rate $\rho$, as shown in Eq (10). The original image is sequentially sliced by moving $x$ from left to right and from top to bottom, dividing the original detection image $I$ into $l$ overlapping blocks $m \times m$ with overlap $P_1^I, P_2^I, \cdots, P_l^I$.

$$\begin{cases} x = w \times \rho \\ S = x \times x \end{cases} \tag{10}$$

Subsequently, while maintaining the aspect ratio, the image scaling operation is completed. A target detector is used to independently perform object detection on each overlapping block after the image scaling operation, yielding the detection results for each overlapping block. Non-Maximum Suppression is applied to merge the detection results of each overlapping block back to the size of the original detection image. In light of the issue of poor recognition accuracy for small foreign objects in the detection task of Pu-erh sun-dried green tea, the improved YOLOv8 detection model proposed earlier is integrated and applied in conjunction with the slice-assisted super inference method during the model inference phase, in order to enhance the accuracy of small foreign object detection in Pu-erh sun-dried green tea.

## Comparative experimental design

To evaluate the improved YOLOv8 model's ability to detect small foreign objects in Pu-erh sun-dried green tea, this study compared it with the original YOLOv8, Faster-RCNN, Corner-Net, and SSD models through extensive experiments. For scientific and reproducible results, a consistent experimental platform and software version were used for training all models. Specific configuration parameters are shown in Table 1.

To thoroughly assess the improved YOLOv8 model's effectiveness in detecting small foreign objects in Pu-erh sun-dried green tea, this study utilized an advanced binary confusion matrix analysis. This method allows for the precise calculation of key classification metrics: Precision [48] and Recall [49]. Additionally, to evaluate model performance comprehensively,

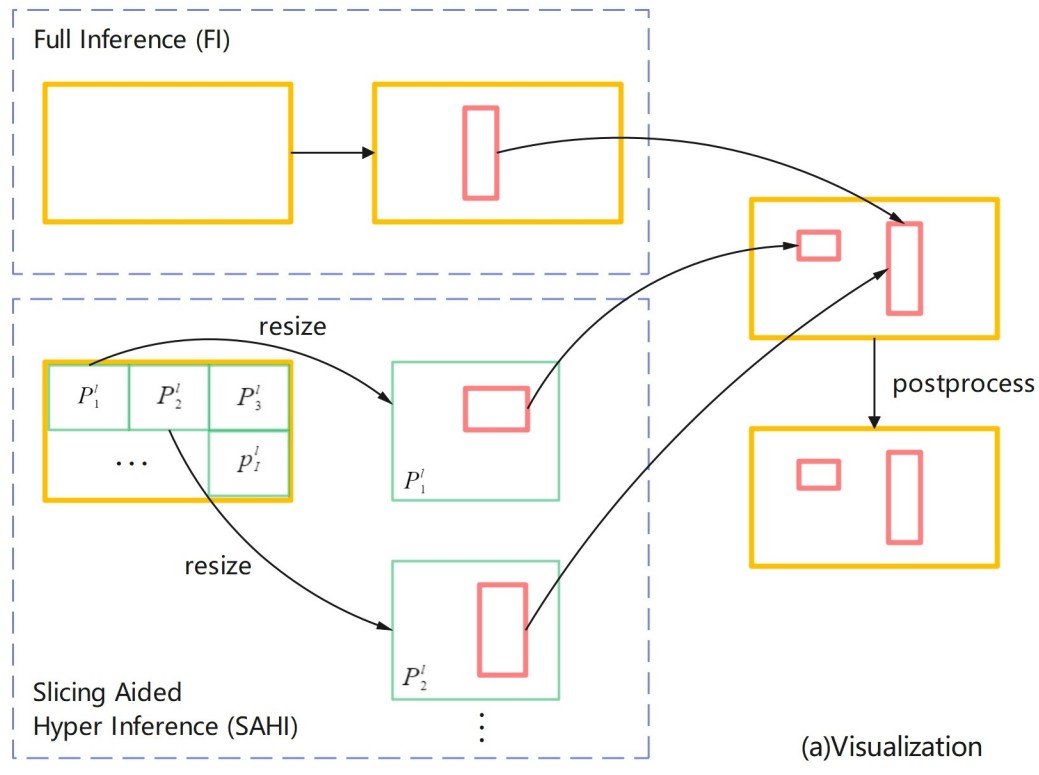

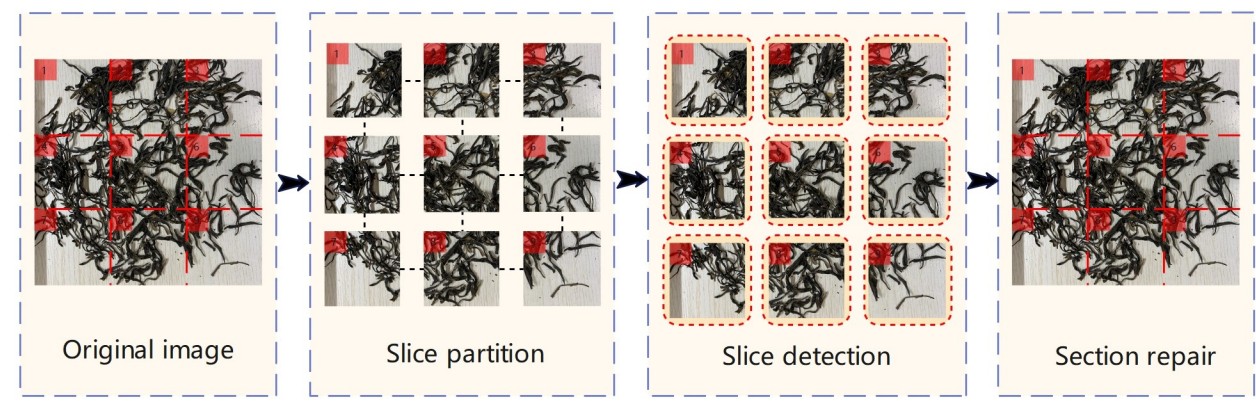

**Fig 12. The principle of slice-assisted super inference.**

the F1 score [50], Average Precision (AP) [51], and mean Average Precision (mAP) [52] were included as metrics. Together, these metrics form an integrated evaluation system to measure the model's classification accuracy and reliability in object detection tasks. The following formulas provide a quantitative method for in-depth analysis of model performance.

$$Precision = \frac{TP}{TP + FP} \times 100\% \qquad (11)$$

**Table 1. Experimental environment configuration and parameter settings.**

| Configuration item | Configuration parameter |
|---|---|
| Operating system | Windows 10 |
| CPU | Intel(R)CORE(TM)i7-11700 |
| Memory | 2933MHz DDR4 ECC |
| Solid state drive | M.2 1TB PCIe NVMe Class 50 |
| GPU | NVIDIA RTX A6000 |
| Compiled language | Python 3.9 |
| Software framework | PyCharm 2019 |
| CUDA | CUDA Version: 12.2 |
| Epochs | 1000 |
| Batch size | 32 |

$$Recall = \frac{TP}{TP + FN} \times 100\% \tag{12}$$

$$F_1 = 2 \times \frac{Precision \times Recall}{Precision + Recall} \tag{13}$$

$$AP = \int_0^1 Precision(Recall)dRecall \tag{14}$$

$$mAP = \frac{1}{M}\sum_{i=1}^{M} AP(i) \tag{15}$$

Where, TP denotes the count of true positives where the model correctly identifies actual foreign objects; FP denotes the count of false positives where the model erroneously classifies non-foreign objects as foreign; FN denotes the count of false negatives where the model fails to detect actual foreign objects.

## Results and analysis

### Ablation study

Based on the original YOLOv8 framework, this study implemented a series of enhancements to improve its performance in detecting small foreign objects in Pu-erh sun-dried green tea. An exhaustive statistical analysis was conducted to systematically verify the effectiveness of these improvements, precisely assessing the individual and combined contributions to model performance. The ablation study comparative data are presented in Table 2.

As demonstrated in Table 2, this study introduced the MPDIoU optimized loss function to enhance the model's sensitivity to target regions, achieving more precise bounding box positioning. Key metrics of Precision, Recall, and mAP improved by 0.83%, 0.78%, and 0.69% respectively over the original YOLOv8. Integration of EfficientDet into the YOLOv8 framework led to model lightweighting, with Parameters and Gradients metrics decreasing by 4.06%, and a 0.6G reduction in model size (GFLOPs). Metrics showed varying improvements, with Precision, Recall, and mAP increasing by 1.86%, 1.01%, and 0.81% respectively, Precision showing the greatest gain. Incorporating BiFormer, with its bidirectional attention mechanism, allowed for the consideration of both forward and backward dependencies in the input

**Table 2. Comparison results of ablation experiments.**

| Model | P (%) | R (%) | mAP (%) | Layers | Parameters | Gradients | GFLOPs |
|---|---|---|---|---|---|---|---|
| YOLOv8 | 90.86 | 90.35 | 94.15 | 225 | 3157200 | 3157184 | 8.9 |
| YOLOv8-M | 91.69 | 91.13 | 94.84 | 225 | 3157200 | 3157184 | 8.9 |
| YOLOv8-E | 92.72 | 91.36 | 94.96 | 239 | 3029175 | 3029159 | 8.3 |
| YOLOv8-B | 93.02 | 90.91 | 95.47 | 236 | 3174480 | 3174464 | 9.5 |
| YOLOv8-ME | 92.09 | 91.59 | 95.23 | 239 | 3029175 | 3029159 | 8.3 |
| YOLOv8-MB | 93.26 | 92.82 | 96.32 | 236 | 3174480 | 3174464 | 9.5 |
| YOLOv8-EB | 93.27 | 92.88 | 96.21 | 250 | 3191816 | 3191816 | 9.6 |
| YOLOv8-MEB | 95.36 | 95.65 | 97.78 | 250 | 3191816 | 3191816 | 9.6 |

Note: M indicates MPDIoU improvement; E indicates EfficientDet enhancement; B indicates BiFormer refinement.

data, capturing context more comprehensively. This enhanced the model's ability to identify small foreign objects in Pu-erh sun-dried green tea, with only a 0.6G increase in model size, but a significant boost in Precision, Recall, and mAP by 2.16%, 0.56%, and 1.32% respectively. Overall, the improved model saw a substantial increase in performance metrics, with a mere 0.7G increase in size, and Precision, Recall, and mAP rose significantly by 4.50%, 5.30%, and 3.63% over the original YOLOv8. Collectively, the YOLOv8-MEB model demonstrated superior performance in detecting small foreign objects in Pu-erh sun-dried green tea from the perspective of smart agricultural devices, offering robust support and new insights for foreign object detection technology in the field of smart agriculture.

This study utilized Gradient-weighted Class Activation Mapping (Grad-CAM), an advanced visualization tool for gaining deeper insights into the model's decision-making process. During the ablation study, gradients were calculated for specific layers of each model to identify regions most influential in image classification decisions. Grad-CAM generates heatmaps that visually reveal key areas in image classification results, highlighted to provide intuitive explanations for model predictions. As shown in Fig 13, the Grad-CAM heatmaps produced by the improved YOLOv8-MEB model matched the actual foreign object areas more closely, indicating the effectiveness of the model's enhancements in recognizing small foreign objects.

## Loss function analysis

The loss function, a critical metric for model performance, primarily measures the discrepancy between predicted and observed values. As shown in Fig 14, the improved YOLOv8-MEB model exhibited a rapid decrease in loss during the initial training phase, indicating good adaptability to the training data. Around the 110th training epoch, the rate of loss reduction began to decelerate, signaling the model's convergence. After 300 epochs, the loss curve stabilized, indicating a stable training state without evident overfitting or significant fluctuations, further confirming the model's stability and robustness.

## Model performance analysis

As depicted in Fig 15, the YOLOv8-MEB model presented in this study achieved performance metrics of 95.36% Precision, 95.65% Recall, and an F1 score of 95.50%. Compared to the original YOLOv8 model, the YOLOv8-MEB model saw a 4.50% increase in Precision, crucial for the detection of small foreign objects in Pu-erh sun-dried green tea due to its direct impact on the accuracy of detection results. Additionally, a 5.30% enhancement in Recall indicates the

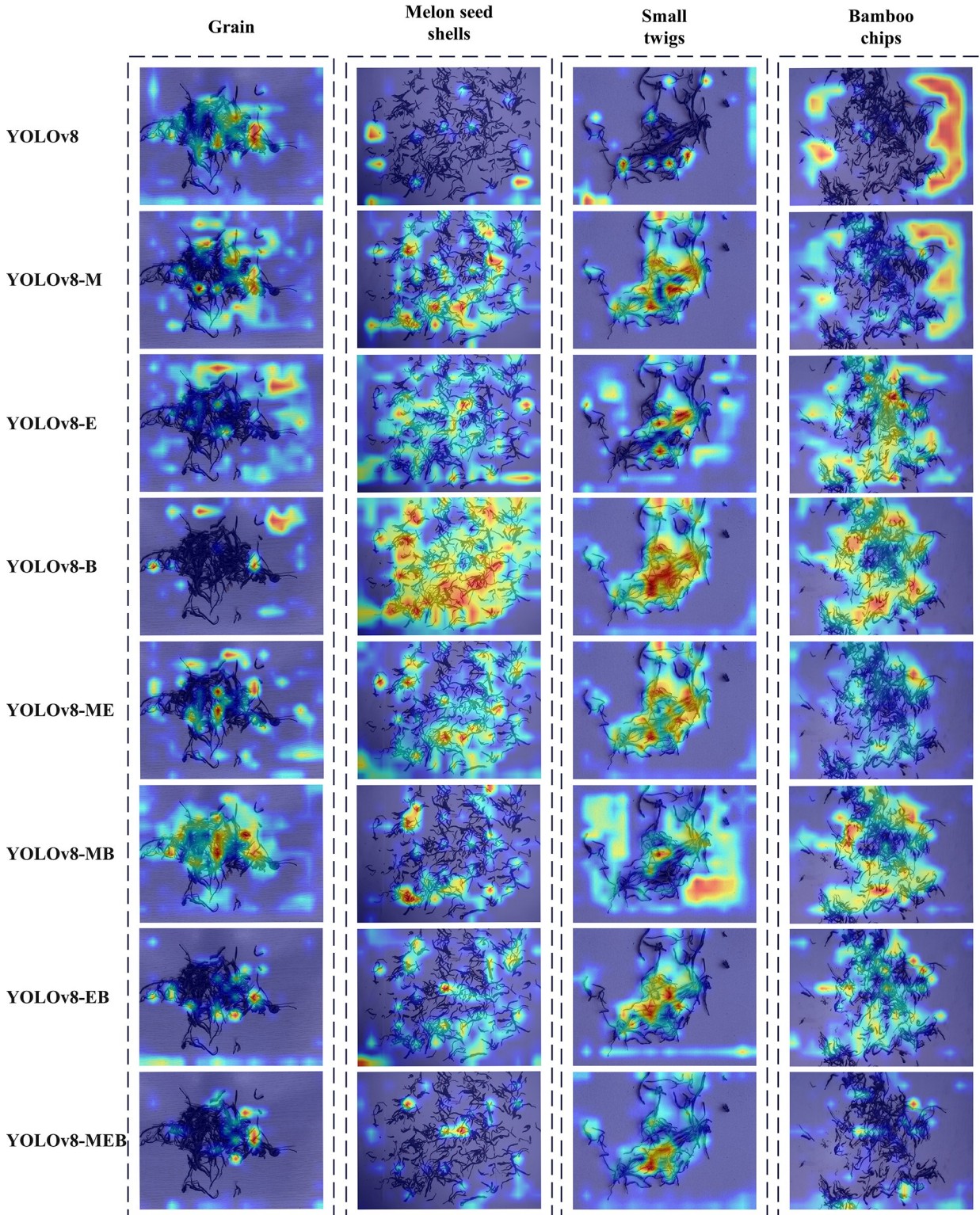

**Fig 13. Grad-CAM thermal maps related to ablation experiments.**

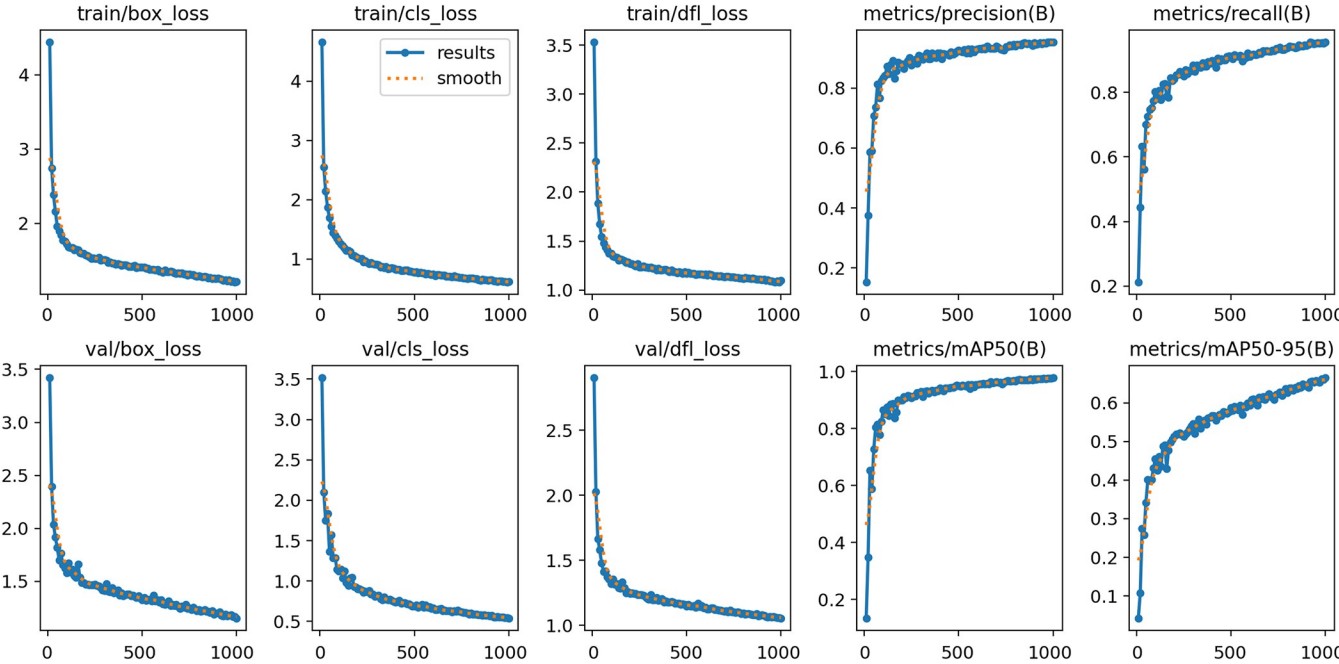

**Fig 14. Change curve of loss function of improved model.**

improved model's increased effectiveness in identifying all present foreign objects. The F1 score also rose by 4.9%, reflecting a better balance between Precision and Recall. In summary, the YOLOv8-MEB model's exceptional ability for efficient and accurate identification of small foreign object targets in Pu-erh sun-dried green tea was significant for quality control of agricultural products like Pu-erh, effectively identifying and detecting small foreign objects to ensure product purity and safety.

## Comparative analysis of different models

This study conducted a comprehensive evaluation of the improved YOLOv8-MEB model, focusing on its accuracy and mean average precision in the detection tasks of four different foreign objects in Pu-erh raw green tea, and conducted an in-depth comparative analysis with advanced network models such as YOLOv8, YOLOv7, YOLOv5, Faster-RCNN, and SSD. Table 3 demonstrates the significant advantages of the YOLOv8-MEB model in foreign object detection.

The improved YOLOv8-MEB model achieved an increase in AP values of 3.60%, 3.89%, 7.17%, 13.97%, and 11.30% respectively for Grain foreign object identification compared to the YOLOv8, YOLOv7, YOLOv5, Faster-RCNN, and SSD network models. For Melon seed shells foreign objects, the AP values increased by 3.64%, 4.04%, 7.23%, 14.10%, and 11.40% respectively. For Small twigs foreign object identification, the AP values increased by 3.78%, 4.06%, 7.49%, 14.24%, and 11.44% respectively. For Bamboo chips foreign object identification, the AP values increased by 3.50%, 3.69%, 7.15%, 13.81%, and 11.06% respectively, with the final mAP increasing by 3.63%, 3.92%, 7.26%, 14.03%, and 11.30% respectively. This indicates that the YOLOv8-MEB network model is more accurate in predicting the location of foreign object targets, performs better in the detection of small target foreign objects in Pu-erh raw green tea across different categories, and has higher objectivity and reliability.

To thoroughly verify the accuracy and efficiency of the optimized YOLOv8-MEB model in identifying common foreign impurities in the grading task of Pu-erh sun-dried green tea, this

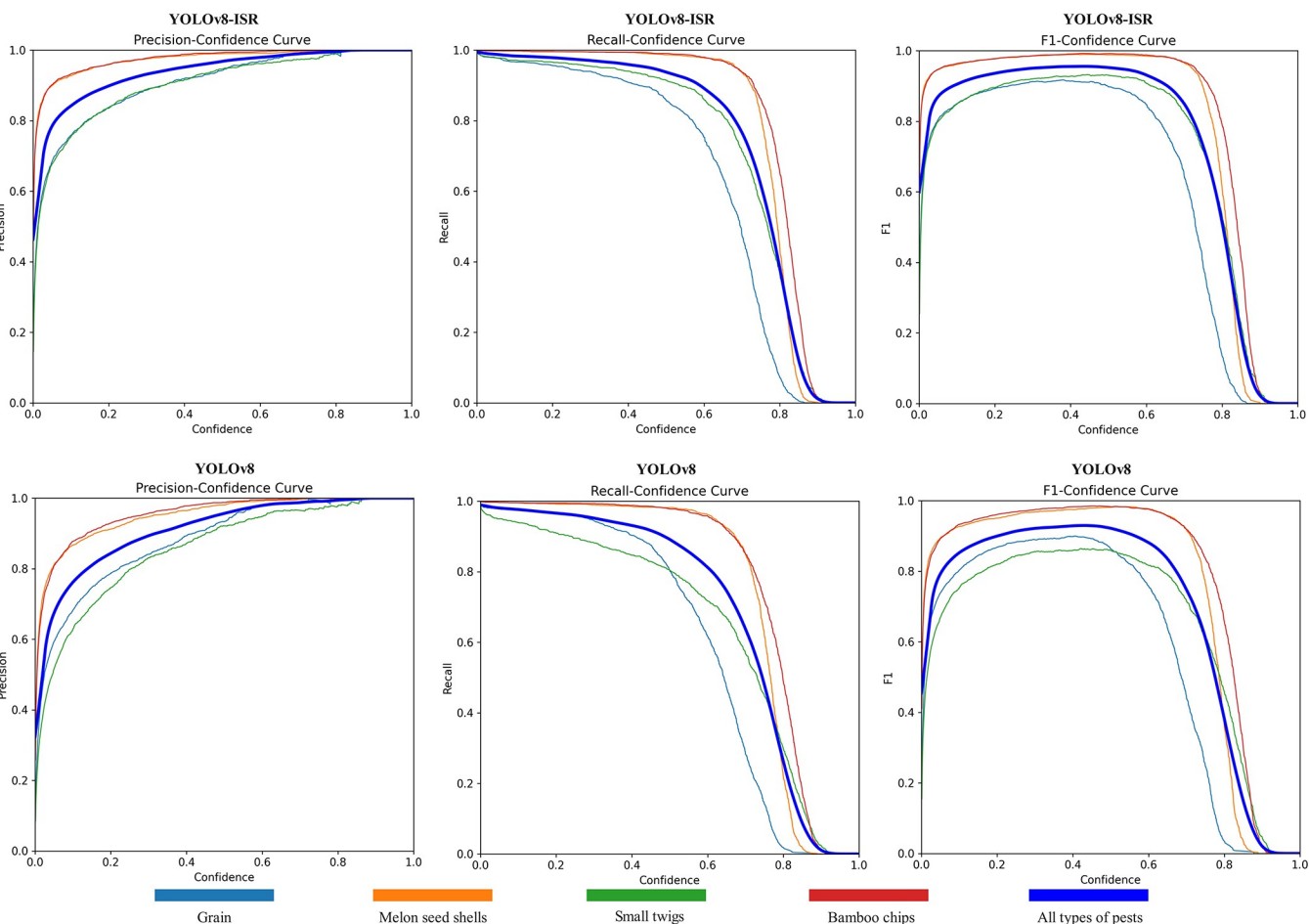

**Fig 15. Curves of Precision, Recall and equilibrium score F1.**

study selected a real dataset from the Tea Plant and Processing Science Observational Experiment Station of the College of Tea Science, Yunnan Agricultural University (25.13° N latitude, 102.75° E longitude) for detailed validation. The dataset includes 50 representative image samples, fully reflecting detection challenges in real scenarios. Under various lighting conditions, the YOLOv8-MEB model demonstrated excellent detection performance in both single and multiple target scenarios. This study conducted a detailed performance comparison with comparative models, as intuitively presented in Fig 16, showing the comparative results. In **Fig 16**,

**Table 3. Comparative experimental data.**

| Model name | AP (Grain) | AP (Melon seed shells) | AP (Small twigs) | AP (Bamboo chips) | mAP |
|---|---|---|---|---|---|
| YOLOv8-MEB | 97.71 | 97.82 | 97.97 | 97.62 | 97.78 |
| YOLOv8 | 94.11 | 94.18 | 94.19 | 94.12 | 94.15 |
| YOLOv7 | 93.82 | 93.78 | 93.91 | 93.93 | 93.86 |
| YOLOv5 | 90.54 | 90.59 | 90.48 | 90.47 | 90.52 |
| Faster-RCNN | 83.74 | 83.72 | 83.73 | 83.81 | 83.75 |
| SSD | 86.41 | 86.42 | 86.53 | 86.56 | 86.48 |

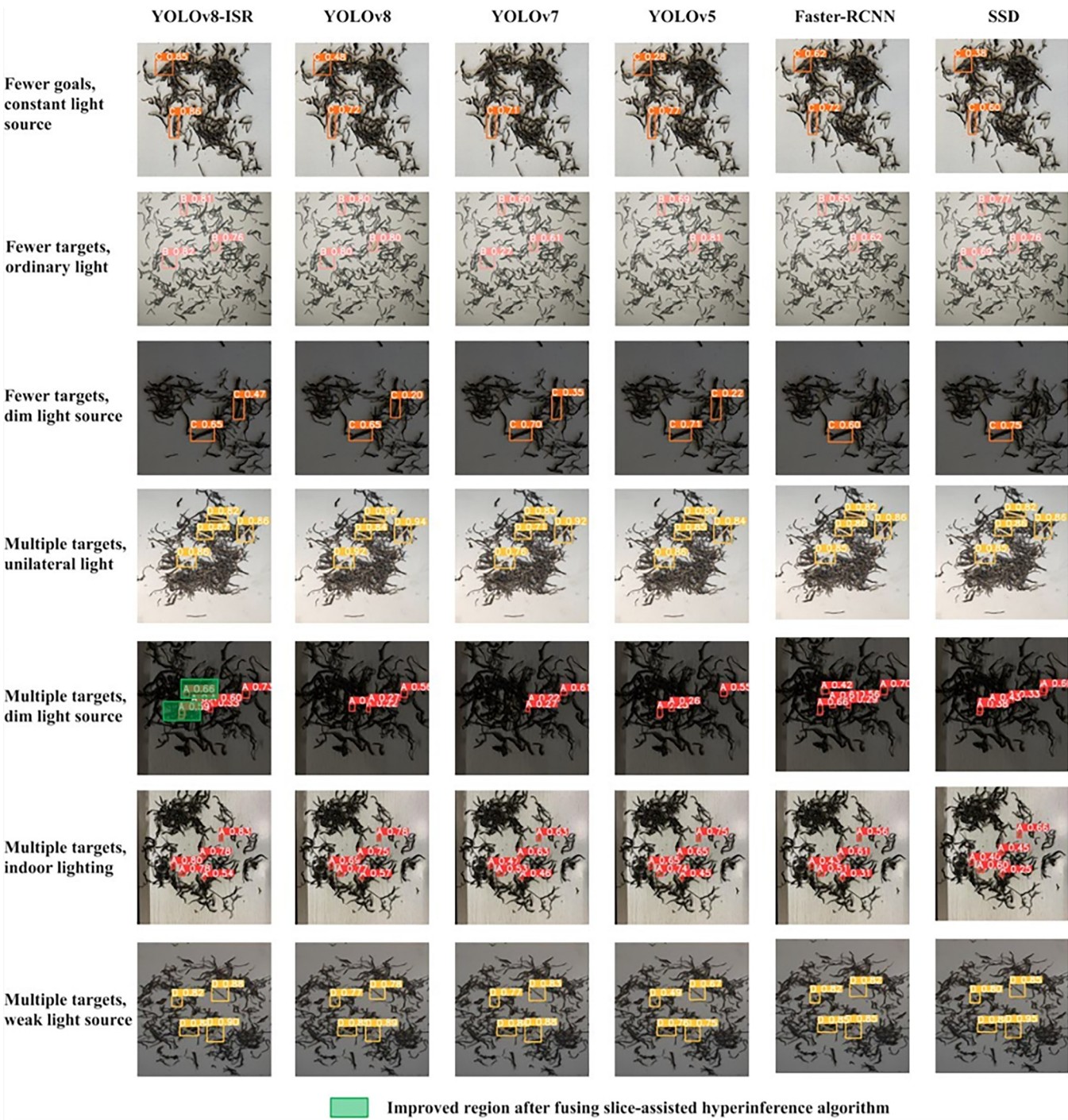

**Fig 16. Comparison of detection results of different models.**

labels A, B, C, and D represent Grain, Melon seed shells, Small twigs, and Bamboo chips, respectively.

The experimental results indicate that under diverse lighting conditions such as constant light sources, ordinary light, dim light, unilateral light, indoor lighting, and weak light sources, there is a significant difference in the foreign object detection capabilities of various target

detection models when the number of targets is low compared to when the number of targets is high. Through in-depth confidence analysis, this study found that the YOLOv8-MEB model demonstrated exceptional confidence levels in both sparse and dense scenes of foreign object targets. The model's confidence was significantly higher in all tested conditions compared to other models such as YOLOv8, YOLOv7, YOLOv5, SSD, and Faster-RCNN, which exhibited relatively lower confidence under the same conditions. The YOLOv8-MEB model's ability to achieve such remarkable high-confidence performance is attributed to its advanced algorithmic architecture and optimization strategies, which not only show significant advantages in the localization accuracy of small foreign objects but also effectively reduce the probability of false positives and repeated detections. Especially in environments with poor lighting and a large number of targets, the performance of the YOLOv8-MEB model is particularly outstanding, accurately identifying all multi-target foreign objects, including complete and partial ones, while maintaining high confidence. In stark contrast, the other comparative models involved in this study, although capable of foreign object detection to some extent, showed a noticeable decline in confidence accuracy and were accompanied by issues of target target recognition omissions. This indicates that in actual application scenarios with complex lighting conditions and variable target numbers, these models lack robustness and cannot meet the requirements for high-precision foreign object detection. A comprehensive assessment of the experimental results shows that the YOLOv8-MEB model in this study demonstrates excellent robustness and reliability in foreign object detection tasks, and its performance is significantly better than other models tested. This provides strong technical support and theoretical basis for efficient and accurate small foreign object detection in Pu-erh raw green tea under various lighting conditions in the future.

## Discussion

This study is dedicated to the detection and identification of foreign objects in the production process of Pu-erh raw green tea, ensuring the quality and hygiene of Pu-erh tea as a food product. To address the issue of rapid detection and precise identification of small foreign objects in Pu-erh raw green tea, this study proposes an improved YOLOv8 network model for foreign object detection. In the field of agriculture, the YOLO algorithm has shown broad potential, especially in defect detection, quality grading, safety detection, and foreign object identification of agricultural products. Li et al. [21] proposed a lightweight improved YOLOv5s model that achieved a precision rate of 97.80% in detecting dragon fruit under daytime and nighttime lighting conditions. In comparison, the improved YOLOv8 model in this study achieved Precision, Recall, mAP, and F1 scores of 95.36%, 95.65%, 97.78%, and 95.50%, respectively, in the task of detecting small foreign objects in Pu-erh raw green tea, demonstrating adaptability in small foreign object identification and detection, with significant improvements in accuracy and evaluation metrics compared to the YOLOv5s model's performance in dragon fruit detection tasks.

In terms of hardware deployment, adaptability and maintenance costs are key considerations in actual production environments. The model proposed in this study has considered lightweight and computational efficiency in its design to adapt to different hardware platforms. Compared to the original model, the improved YOLOv8 network model in this study has reduced Parameters and Gradients indicators by 1.10%, while significantly increasing Precision, Recall, mAP, and F1 by 4.50%, 5.30%, 3.63%, and 4.9%, respectively. This indicates that the model has optimized the use of computational resources while maintaining high accuracy. Compared to the Mask-YOLOv7 model based on spatio-temporal convolutional neural networks proposed by Bello et al. [22], which achieved detection accuracy rates of 93% and 95%

in controlled and uncontrolled environments, respectively, the model in this study has improved the accuracy rate to 97.78% in the detection of small foreign objects in Pu-erh raw green tea, demonstrating higher detection accuracy and robustness.

Furthermore, the scalability and real-time application of the model are also a focus of this study. The improved YOLOv8 model in this study has been optimized in terms of computational efficiency and resource requirements to support large-scale industrial use and real-time applications. Compared to the unmanned pineapple harvesting model proposed by Meng et al. [23], which achieved a detection accuracy rate of only 92.54% in identifying pineapples, the model in this study has a higher accuracy rate in the detection of small foreign objects in Pu-erh raw green tea, which is crucial for real-time detection and sorting systems. Although the size of the improved model in this study has increased by 7.87% compared to the original model, this increase is mainly attributed to the increase in model complexity to improve detection accuracy and robustness. This study has achieved a breakthrough in foreign object detection technology for Pu-erh raw green tea, laying the foundation for the automation and intelligence of the tea industry. By improving detection accuracy and speed, this technology has optimized the traditional tea production process, reduced human errors, increased efficiency, and ensured product quality and safety. The innovativeness of the improved YOLOv8 network model provides an efficient solution for foreign object detection in the agricultural and food processing industries, which is expected to enhance industry standards and consumer confidence. The potential of the technology in real-time monitoring and data processing is enormous and will support food safety regulation.

In future research, the aim is to construct an efficient foreign object detection and identification screening device for Pu-erh raw green tea. The research will focus on the lightweight of the software algorithm model and the optimization of the hardware framework design. On the software side, model compression techniques such as pruning, quantization, and knowledge distillation will be used to reduce the model size while maintaining detection performance. At the same time, the algorithm will be optimized for cross-platform compatibility and real-time performance to ensure flexibility in operation on different hardware and low-latency detection in production environments. In the design and deployment of the hardware framework, modular and integrated strategies will be adopted to improve the maintainability and scalability of the equipment, ensuring its adaptability to diverse production environments. Through highly integrated component design, the aim is to achieve smoothness and efficiency in the operation process, while reducing spatial occupancy and improving energy efficiency. In addition, intuitive user interfaces and automated calibration mechanisms will be developed to enhance the flexibility and operability of the system. These designs will allow operators to easily configure and monitor the system, reducing manual intervention and improving the system's adaptability and accuracy. At the same time, integrated remote monitoring and control functions will be added to allow monitoring and control of the equipment from different locations, further enhancing the management efficiency of the equipment.

## Conclusion

This study proposes an improved YOLOv8 network model-based foreign object detection method aimed at enhancing the rapid detection and precise identification of small foreign objects in the production process of Pu-erh raw green tea, ensuring the quality and hygiene safety of Pu-erh tea. The improved YOLOv8 model in this study employs an MPDIoU optimized loss function to improve target detection accuracy, integrates the EfficientDet architecture to enhance detection efficiency for targets of different sizes, and utilizes the BiFormer bidirectional attention mechanism to enhance the understanding of image context.

Additionally, the introduction of slice-assisted super reasoning technology further improves the model's recognition accuracy and robustness for small targets and multi-scale foreign objects, ensuring the quality and hygiene safety of Pu-erh raw green tea.

Compared to the original model, the improved YOLOv8 model in this study has significantly increased Precision, Recall, mAP, and F1 by 4.50%, 5.30%, 3.63%, and 4.9%, respectively, while also reducing the Parameters and Gradients indicator parameters by 1.10%. Moreover, compared to the original model, YOLOv7, YOLOv5, Faster-RCNN, and SSD network models, the AP values for Grain foreign object identification have increased by 3.60%, 3.89%, 7.17%, 13.97%, and 11.30%, respectively. For Melon seed shells foreign objects, the AP values have increased by 3.64%, 4.04%, 7.23%, 14.10%, and 11.40%, respectively. For Small twigs foreign object identification, the AP values have increased by 3.78%, 4.06%, 7.49%, 14.24%, and 11.44%, respectively. For Bamboo chips foreign object identification, the AP values have increased by 3.50%, 3.69%, 7.15%, 13.81%, and 11.06%, respectively, with the final mAP increasing by 3.63%, 3.92%, 7.26%, 14.03%, and 11.30%, respectively. The improved model in this study possesses efficient foreign object recognition capabilities, providing technical support for intelligent detection technology on the foreign object sorting line of Pu-erh raw green tea, and also laying the foundation for the smartization of the tea industry and the enhancement of tea quality.

## Acknowledgments

The author would like to thank the anonymous reviewers for their careful review of the paper and their kind suggestions to improve the overall quality of the manuscript.

## Author Contributions

**Conceptualization:** Houqiao Wang, Zejun Wang.

**Data curation:** Houqiao Wang, Shihao Zhang, Qiang Zhao.

**Funding acquisition:** Zejun Wang.

**Investigation:** Houqiao Wang, Xiaoxue Guo, Zejun Wang.

**Project administration:** Houqiao Wang, Xiaoxue Guo, Gongming Li, Qiang Zhao.

**Resources:** Houqiao Wang, Xiaoxue Guo, Shihao Zhang.

**Software:** Shihao Zhang, Zejun Wang.

**Supervision:** Houqiao Wang, Gongming Li, Qiang Zhao, Zejun Wang.

**Validation:** Shihao Zhang, Gongming Li.

**Writing – original draft:** Houqiao Wang, Qiang Zhao.

**Writing – review & editing:** Gongming Li, Qiang Zhao.

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
