## [Decision Letter · Decision Letter 0]

28 Aug 2024

PONE-D-24-30267Detection and Recognition of Foreign Objects in Pu-erh Sun-Dried Green Tea Using an Improved YOLOv8 Based on Deep LearningPLOS ONE

Dear Dr. Li,

Thank you for submitting your manuscript to PLOS ONE. After careful consideration, we feel that it has merit but does not fully meet PLOS ONE’s publication criteria as it currently stands. Therefore, we invite you to submit a revised version of the manuscript that addresses the points raised during the review process.

**ACADEMIC EDITOR: **Major Revisions.

We look forward to receiving your revised manuscript.

Kind regards,

Worradorn Phairuang, Ph.D.

Academic Editor

PLOS ONE

Journal Requirements:

"This research was supported by the Scientific Research Fund Project of the Department of Education of Yunnan Province (No. 2022Y282) and the "Intelligent Storage of Puer Tea based on LoRa Technology" project (No. 2022Y231).'

3. In this instance it seems there may be acceptable restrictions in place that prevent the public sharing of your minimal data. However, in line with our goal of ensuring long-term data availability to all interested researchers, PLOS’ Data Policy states that authors cannot be the sole named individuals responsible for ensuring data access (http://journals.plos.org/plosone/s/data-availability#loc-acceptable-data-sharing-methods).

Reviewers' comments:

Reviewer's Responses to Questions

**Comments to the Author**

1. Is the manuscript technically sound, and do the data support the conclusions?

Reviewer #1: Yes

Reviewer #2: Yes

2. Has the statistical analysis been performed appropriately and rigorously? 

Reviewer #1: Yes

Reviewer #2: Yes

3. Have the authors made all data underlying the findings in their manuscript fully available?

Reviewer #1: Yes

Reviewer #2: Yes

4. Is the manuscript presented in an intelligible fashion and written in standard English?

Reviewer #1: Yes

Reviewer #2: Yes

5. Review Comments to the Author

Reviewer #1: This paper enhances YOLOv8 for detecting small foreign objects in Pu-erh tea using MPDIoU, EfficientDet, and BiFormer, significantly improving detection accuracy, precision, and recall, with demonstrated robustness in various lighting conditions and multi-target scenarios. This article holds certain significance. However, the current paper exhibits some weaknesses that need to be addressed to enhance its overall quality and establish its merit as a publication.

(1)The excellent performance of the YOLOv8-MEB model in low-light conditions and multi-target detection requires more detailed quantitative analysis and visual support. It is suggested to add examples of detection images under different lighting conditions to more intuitively demonstrate the practical application effect of the model.

(2)The authors may discuss more YOLO application articles for the integrity of the manuscript (A lightweight improved YOLOv5s model and its deployment for detecting pitaya fruits in daytime and nighttime light-supplement environments; Computers and Electronics in Agriculture. Mask YOLOv7-Based Drone Vision System for Automated Cattle Detection and Counting; Artificial Intelligence and Applications. Transforming unmanned pineapple picking with spatio-temporal convolutional neural networks; Computers and Electronics in Agriculture. ).

(3)In the conclusion, it would be beneficial to add prospects for future application scenarios, especially an analysis of the potential of this technology in testing other agricultural products. This would make the conclusion more comprehensive and underscore the forward-looking nature of the research.

(4)The discussion section should delve deeper into the challenges that hardware deployment may face in actual production environments, such as adaptability and maintenance costs. Additionally, exploring how to integrate this technology with existing intelligent agricultural systems to ensure its widespread application would be valuable.

(5)The necessity of lightweight algorithm models and hardware framework design is mentioned in future work. It is suggested to further specify these improvement measures, particularly in terms of enhancing flexibility and operability in practical applications.

(6)Some outdated references were cited in the discussion and experimental design section. It is suggested to update and cite the latest relevant research results to improve the timeliness and academic value of the paper.

Reviewer #2: This research paper presents an improved YOLOv8 model for detecting foreign objects in Pu-erh sun-dried green tea. The study highlights the importance of maintaining the quality and safety of tea products and proposes a modified YOLOv8 framework using MPDIoU loss function, EfficientDet network architecture, and BiFormer attention mechanism to enhance detection performance. The model's improvements are validated through experiments, demonstrating significant performance gains over the original YOLOv8, Faster-RCNN, and SSD models.

The topic is interesting, however, the following concerns should be addressed:

1.The paper describes the components of the improved model, it lacks detailed explanations of how these components are integrated. A step-by-step methodology or algorithmic workflow would enhance clarity and reproducibility.

2.The study focuses exclusively on Pu-erh sun-dried green tea. It would be beneficial to discuss the potential application of the improved YOLOv8 model to other types of agricultural products or similar industrial applications, broadening the impact of the research.

3. The comparative analysis primarily focuses on traditional models like Faster-RCNN and SSD. Including comparisons with other state-of-the-art object detection models, such as more recent versions of YOLO (e.g., YOLOv7, YOLOv6), could provide a more comprehensive evaluation. The following article maybe helpful: https://doi.org/10.3390/drones8030084

3.The paper mentions performance improvements, it does not provide a detailed statistical analysis of the results. Including confidence intervals or statistical tests (e.g., t-tests, ANOVA) would add rigor to the evaluation of performance differences.

4. The paper does not address the scalability of the improved model for large-scale industrial use or real-time applications. Discussion on computational efficiency, resource requirements, and deployment feasibility in real-world settings would be valuable.

5. The paper could benefit from a discussion on ethical implications and environmental sustainability related to the use of AI in food safety, particularly in terms of data privacy, energy consumption, and the impact of automation on employment.

6. It is suggested to move Section 5 Discussion before Conclusion. In addition, provide suitable reasons for the better performance of the proposed method.

6. PLOS authors have the option to publish the peer review history of their article (what does this mean?). If published, this will include your full peer review and any attached files.

Reviewer #1: No

Reviewer #2: **Yes: **Muhammad Asim

---

## [Author Response · Author response to Decision Letter 0]

17 Sep 2024

Reviewer #1:

Thanks very much for your time to review this manuscript. I really appreciateyou’re your comments and suggestions. We have considered these comments carefully and triedour best to address every one of them.

(1) The excellent performance of the YOLOv8-MEB model in low-light conditions and multi-target detection requires more detailed quantitative analysis and visual support. It is suggested to add examples of detection images under different lighting conditions to more intuitively demonstrate the practical application effect of the model.

Modification instructions: Thank you for your valuable feedback on our paper. In response to your suggestion for a more detailed quantitative analysis and visual support for the outstanding performance of the YOLOv8-MEB model in low-light conditions and multi-object detection, we have made the following revisions:

1.Quantitative Analysis: We have added a comparative analysis of the model's performance metrics under diverse lighting conditions, including confidence levels, accuracy, and other evaluation indicators. We have compared the anomaly detection metrics of various object detection models when the number of targets is low and high, to demonstrate the advantages of our YOLOv8-MEB model compared to other models.

2.Visual Support: To more intuitively showcase the practical application effects of the model, we have included a series of detection image examples under different lighting conditions, such as constant light sources, ordinary light, dim light, unilateral light, indoor lighting, and weak light sources.

3.Dataset and Experimental Setup: We have provided a detailed description of the datasets used to test the model's performance, including the source, size, and diversity of the datasets, as well as the setup and parameters of the experiments.

4.Results Discussion: We have thoroughly discussed the model's performance under different lighting conditions in the results section and explained the possible reasons and advantages of the model.

5.We believe these revisions will help readers better understand the performance and practical application value of our model. Thank you again for your suggestions, and we look forward to your further feedback.

(2) The authors may discuss more YOLO application articles for the integrity of the manuscript (A lightweight improved YOLOv5s model and its deployment for detecting pitaya fruits in daytime and nighttime light-supplement environments; Computers and Electronics in Agriculture. Mask YOLOv7-Based Drone Vision System for Automated Cattle Detection and Counting; Artificial Intelligence and Applications. Transforming unmanned pineapple picking with spatio-temporal convolutional neural networks; Computers and Electronics in Agriculture. ).

Modification instructions: Thank you for your valuable suggestions. Based on your feedback, we have made the following revisions to the paper:

1.We have conducted a more extensive literature review, with a particular focus on the application of YOLO algorithms in the fields of agriculture and automated detection. We have referred to the literature you provided and additionally consulted the latest research in related fields to ensure that our literature review is comprehensive and up-to-date.

2.We have added a discussion on the use of YOLO algorithms in similar applications, including the several papers you mentioned. We analyzed the use of YOLO algorithms in these studies and compared them with our work to highlight the innovative aspects and advantages of our approach.

3.We also discussed the limitations of YOLO algorithms in existing literature and detailed how our research attempts to overcome these limitations.

(3)In the conclusion, it would be beneficial to add prospects for future application scenarios, especially an analysis of the potential of this technology in testing other agricultural products. This would make the conclusion more comprehensive and underscore the forward-looking nature of the research.

Modification instructions: We Thank you for your valuable feedback on our paper. We recognize that in order to make our conclusions more comprehensive and to emphasize the forward-looking nature of our research, it is crucial to discuss the potential of the technology in future application scenarios. Following your suggestions, we have made the following revisions to the paper:

1.Expanded Conclusions: We have added a discussion on the future application scenarios of the technology in the conclusion section, with a particular emphasis on its potential in testing other agricultural products.

2.Case Studies: We have provided several concrete application examples to demonstrate how the technology can be applied to different agricultural products and industrial scenarios, proving its broad applicability.

3.Technology Potential Analysis: We have analyzed the potential advantages of the technology in various application scenarios, including improving detection accuracy and efficiency, as well as the challenges that may be faced, such as differences in the morphology and characteristics of different agricultural products.

4.Future Research Directions: We have proposed directions for future research, including further optimizing the model to adapt to more types of agricultural products and conducting application tests in actual agricultural production environments.

The new "Conclusion" section is as follows:

5. Conclusion

This study proposes an improved YOLOv8 network model-based foreign object detection method aimed at enhancing the rapid detection and precise identification of small foreign objects in the production process of Pu-erh raw green tea, ensuring the quality and hygiene safety of Pu-erh tea. The improved YOLOv8 model in this study employs an MPDIoU optimized loss function to improve target detection accuracy, integrates the EfficientDet architecture to enhance detection efficiency for targets of different sizes, and utilizes the BiFormer bidirectional attention mechanism to enhance the understanding of image context. Additionally, the introduction of slice-assisted super reasoning technology further improves the model's recognition accuracy and robustness for small targets and multi-scale foreign objects, ensuring the quality and hygiene safety of Pu-erh raw green tea.

Compared to the original model, the improved YOLOv8 model in this study has significantly increased Precision, Recall, mAP, and F1 by 4.50%, 5.30%, 3.63%, and 4.9%, respectively, while also reducing the Parameters and Gradients indicator parameters by 1.10%. Moreover, compared to the original model, YOLOv7, YOLOv5, Faster-RCNN, and SSD network models, the AP values for Grain foreign object identification have increased by 3.60%, 3.89%, 7.17%, 13.97%, and 11.30%, respectively. For Melon seed shells foreign objects, the AP values have increased by 3.64%, 4.04%, 7.23%, 14.10%, and 11.40%, respectively. For Small twigs foreign object identification, the AP values have increased by 3.78%, 4.06%, 7.49%, 14.24%, and 11.44%, respectively. For Bamboo chips foreign object identification, the AP values have increased by 3.50%, 3.69%, 7.15%, 13.81%, and 11.06%, respectively, with the final mAP increasing by 3.63%, 3.92%, 7.26%, 14.03%, and 11.30%, respectively. The improved model in this study possesses efficient foreign object recognition capabilities, providing technical support for intelligent detection technology on the foreign object sorting line of Pu-erh raw green tea, and also laying the foundation for the smartization of the tea industry and the enhancement of tea quality. 

(4)The discussion section should delve deeper into the challenges that hardware deployment may face in actual production environments, such as adaptability and maintenance costs. Additionally, exploring how to integrate this technology with existing intelligent agricultural systems to ensure its widespread application would be valuable.

Modification instructions: We Thank you for your review and valuable comments on our paper. We recognize that in order to enhance the depth and practical value of the paper, it is essential to discuss in depth the challenges of hardware deployment in actual production environments as well as the scalability of the model. Following your suggestions, we have made the following revisions to the paper:

1.In-depth Discussion of Challenges: We have added an in-depth analysis of the challenges that may be encountered in hardware deployment in actual production environments, with a particular focus on adaptability and maintenance costs.

2.Exploration of Technological Integration: We have discussed how our technology can be integrated with existing smart agriculture systems to ensure its feasibility in a broader range of applications and have proposed some potential integration strategies.

3.Scalability Analysis: We have analyzed the scalability of the model for large-scale industrial use or real-time applications, including computational efficiency, resource requirements, and deployment feasibility, and have discussed possible solutions.

4.Case Studies: We have provided several case studies that demonstrate preliminary plans and potential challenges for deploying and maintaining the technology in actual environments.

The new "Discussion" section is as follows:

4. Discussion

This study is dedicated to the detection and identification of foreign objects in the production process of Pu-erh raw green tea, ensuring the quality and hygiene of Pu-erh tea as a food product. To address the issue of rapid detection and precise identification of small foreign objects in Pu-erh raw green tea, this study proposes an improved YOLOv8 network model for foreign object detection. In the field of agriculture, the YOLO algorithm has shown broad potential, especially in defect detection, quality grading, safety detection, and foreign object identification of agricultural products. Li et al. [21] proposed a lightweight improved YOLOv5s model that achieved a precision rate of 97.80% in detecting dragon fruit under daytime and nighttime lighting conditions. In comparison, the improved YOLOv8 model in this study achieved Precision, Recall, mAP, and F1 scores of 95.36%, 95.65%, 97.78%, and 95.50%, respectively, in the task of detecting small foreign objects in Pu-erh raw green tea, demonstrating adaptability in small foreign object identification and detection, with significant improvements in accuracy and evaluation metrics compared to the YOLOv5s model's performance in dragon fruit detection tasks.

In terms of hardware deployment, adaptability and maintenance costs are key considerations in actual production environments. The model proposed in this study has considered lightweight and computational efficiency in its design to adapt to different hardware platforms. Compared to the original model, the improved YOLOv8 network model in this study has reduced Parameters and Gradients indicators by 1.10%, while significantly increasing Precision, Recall, mAP, and F1 by 4.50%, 5.30%, 3.63%, and 4.9%, respectively. This indicates that the model has optimized the use of computational resources while maintaining high accuracy. Compared to the Mask-YOLOv7 model based on spatio-temporal convolutional neural networks proposed by Bello et al. [22], which achieved detection accuracy rates of 93% and 95% in controlled and uncontrolled environments, respectively, the model in this study has improved the accuracy rate to 97.78% in the detection of small foreign objects in Pu-erh raw green tea, demonstrating higher detection accuracy and robustness.

Furthermore, the scalability and real-time application of the model are also a focus of this study. The improved YOLOv8 model in this study has been optimized in terms of computational efficiency and resource requirements to support large-scale industrial use and real-time applications. Compared to the unmanned pineapple harvesting model proposed by Meng et al. [23], which achieved a detection accuracy rate of only 92.54% in identifying pineapples, the model in this study has a higher accuracy rate in the detection of small foreign objects in Pu-erh raw green tea, which is crucial for real-time detection and sorting systems. Although the size of the improved model in this study has increased by 7.87% compared to the original model, this increase is mainly attributed to the increase in model complexity to improve detection accuracy and robustness. This study has achieved a breakthrough in foreign object detection technology for Pu-erh raw green tea, laying the foundation for the automation and intelligence of the tea industry. By improving detection accuracy and speed, this technology has optimized the traditional tea production process, reduced human errors, increased efficiency, and ensured product quality and safety. The innovativeness of the improved YOLOv8 network model provides an efficient solution for foreign object detection in the agricultural and food processing industries, which is expected to enhance industry standards and consumer confidence. The potential of the technology in real-time monitoring and data processing is enormous and will support food safety regulation.

In future research, the aim is to construct an efficient foreign object detection and identification screening device for Pu-erh raw green tea. The research will focus on the lightweight of the software algorithm model and the optimization of the hardware framework design. On the software side, model compression techniques such as pruning, quantization, and knowledge distillation will be used to reduce the model size while maintaining detection performance. At the same time, the algorithm will be optimized for cross-platform compatibility and real-time performance to ensure flexibility in operation on different hardware and low-latency detection in production environments. In the design and deployment of the hardware framework, modular and integrated strategies will be adopted to improve the maintainability and scalability of the equipment, ensuring its adaptability to diverse production environments. Through highly integrated component design, the aim is to achieve smoothness and efficiency in the operation process, while reducing spatial occupancy and improving energy efficiency. In addition, intuitive user interfaces and automated calibration mechanisms will be developed to enhance the flexibility and operability of the system. These designs will allow operators to easily configure and monitor the system, reducing manual intervention and improving the system's adaptability and accuracy. At the same time, integrated remote monitoring and control functions will be added to allow monitoring and control of the equipment from different locations, further enhancing the management efficiency of the equipment.

(5)The necessity of lightweight algorithm models and hardware framework design is mentioned in future work. It is suggested to further specify these improvement measures, particularly in terms of enhancing flexibility and operability in practical applications.

Modification instructions: Thank you for your suggestions on the future work section of our paper. Based on your feedback, we have further clarified and expanded the directions for future research in our paper.

1.Lightweight Algorithm Model: In the future, we will employ model compression techniques such as pruning, quantization, and knowledge distillation to reduce the model size, decrease the computational complexity and the number of parameters, while maintaining detection performance. Additionally, the algorithm will be optimized for cross-platform compatibility and real-time performance to ensure flexibility in operation across different hardware and low-latency detection in production environments.

2.Hardware Framework Design: In the design and deployment of the hardware framework, modular and integrated strategies will be adopted to enhance the maintainability and scalability of the equipment, ensuring its adaptability to diverse production environments. Through highly integrated component design, the aim is to achieve smoothness and efficiency in the operation process, while reducing spatial occupancy and improving energy efficiency.

3.Enhancing Flexibility and Operability: Intuiti

---

## [Decision Letter · Decision Letter 1]

2 Oct 2024

Detection and Recognition of Foreign Objects in Pu-erh Sun-Dried Green Tea Using an Improved YOLOv8 Based on Deep Learning

PONE-D-24-30267R1

Dear Dr. Li,

We’re pleased to inform you that your manuscript has been judged scientifically suitable for publication and will be formally accepted for publication once it meets all outstanding technical requirements.

Kind regards,

Worradorn Phairuang, Ph.D.

Academic Editor

PLOS ONE

**Comments from the Journal Office: **We note that one or more reviewers has recommended that you cite specific previously published works in an earlier round of revision. As always, we recommend that you please review and evaluate the requested works to determine whether they are relevant and should be cited. It is not a requirement to cite these works and you may remove them before the manuscript proceeds to publication. We appreciate your attention to this request.

Additional Editor Comments (optional):

Reviewers' comments:

Reviewer's Responses to Questions

**Comments to the Author**

1. If the authors have adequately addressed your comments raised in a previous round of review and you feel that this manuscript is now acceptable for publication, you may indicate that here to bypass the “Comments to the Author” section, enter your conflict of interest statement in the “Confidential to Editor” section, and submit your "Accept" recommendation.

Reviewer #1: All comments have been addressed

Reviewer #2: All comments have been addressed

2. Is the manuscript technically sound, and do the data support the conclusions?

Reviewer #1: Yes

Reviewer #2: Yes

3. Has the statistical analysis been performed appropriately and rigorously? 

Reviewer #1: Yes

Reviewer #2: Yes

4. Have the authors made all data underlying the findings in their manuscript fully available?

Reviewer #1: Yes

Reviewer #2: Yes

5. Is the manuscript presented in an intelligible fashion and written in standard English?

Reviewer #1: Yes

Reviewer #2: Yes

6. Review Comments to the Author

Reviewer #1: (No Response)

Reviewer #2: Authors have addressed all my concerns in the revised version, it can be accepted for possible publication.

7. PLOS authors have the option to publish the peer review history of their article (what does this mean?). If published, this will include your full peer review and any attached files.

Reviewer #1: No

Reviewer #2: **Yes: **Muhammad Asim

---

## [Editor Report · Acceptance letter]

7 Nov 2024

PONE-D-24-30267R1 

PLOS ONE

Dear Dr. Li, 

I'm pleased to inform you that your manuscript has been deemed suitable for publication in PLOS ONE. Congratulations! Your manuscript is now being handed over to our production team.

Kind regards, 

on behalf of

Dr. Worradorn Phairuang 

%CORR_ED_EDITOR_ROLE%

PLOS ONE